# A Comparative Framework for Preconditioned Lasso Algorithms

**Fabian L. Wauthier**
Statistics and WTCHG
University of Oxford
flw@stats.ox.ac.uk

**Nebojsa Jojic**
Microsoft Research, Redmond
jojic@microsoft.com

**Michael I. Jordan**
Computer Science Division
University of California, Berkeley
jordan@cs.berkeley.edu

## Abstract

The Lasso is a cornerstone of modern multivariate data analysis, yet its performance suffers in the common situation in which covariates are correlated. This limitation has led to a growing number of *Preconditioned Lasso* algorithms that pre-multiply $X$ and $y$ by matrices $P_X$, $P_y$ prior to running the standard Lasso. A direct comparison of these and similar Lasso-style algorithms to the original Lasso is difficult because the performance of all of these methods depends critically on an auxiliary penalty parameter $\lambda$. In this paper we propose an agnostic framework for comparing Preconditioned Lasso algorithms to the Lasso without having to choose $\lambda$. We apply our framework to three Preconditioned Lasso instances and highlight cases when they will outperform the Lasso. Additionally, our theory reveals fragilities of these algorithms to which we provide partial solutions.

## 1 Introduction

Variable selection is a core inferential problem in a multitude of statistical analyses. Confronted with a large number of (potentially) predictive variables, the goal is to select a small subset of variables that can be used to construct a parsimonious model. Variable selection is especially relevant in linear observation models of the form

$$y = X\beta^* + w \quad \text{with} \quad w \sim \mathcal{N}(0, \sigma^2 I_{n \times n}), \tag{1}$$

where $X$ is an $n \times p$ matrix of features or predictors, $\beta^*$ is an unknown $p$-dimensional regression parameter, and $w$ is a noise vector. In high-dimensional settings where $n \ll p$, ordinary least squares is generally inappropriate. Assuming that $\beta^*$ is sparse (i.e., the support set $S(\beta^*) \triangleq \{i | \beta_i^* \neq 0\}$ has cardinality $k < n$), a mainstay algorithm for such settings is the Lasso [10]:

$$\text{Lasso:} \quad \hat{\beta} = \operatorname{argmin}_{\beta \in \mathbb{R}^p} \frac{1}{2n} \|y - X\beta\|_2^2 + \lambda \|\beta\|_1. \tag{2}$$

For a particular choice of $\lambda$, the variable selection properties of the Lasso can be analyzed by quantifying how well the estimated support $S(\hat{\beta})$ approximates the true support $S(\beta^*)$. More careful analyses focus instead on recovering the *signed* support $S_\pm(\beta^*)$,

$$S_\pm(\beta_i^*) \triangleq \begin{cases} +1 & \text{if } \beta_i^* > 0 \\ -1 & \text{if } \beta_i^* < 0 \\ 0 & \text{o.w.} \end{cases}. \tag{3}$$

Theoretical developments during the last decade have shed light onto the support recovery properties of the Lasso and highlighted practical difficulties when the columns of $X$ are correlated. These developments have led to various conditions on $X$ for support recovery, such as the *mutual incoherence* or the *irrepresentable condition* [1, 3, 8, 12, 13].

In recent years, several modifications of the standard Lasso have been proposed to improve its support recovery properties [2, 7, 14, 15]. In this paper we focus on a class of "Preconditioned Lasso" algorithms [5, 6, 9] that pre-multiply $X$ and $y$ by suitable matrices $P_X$ and $P_y$ to yield $\bar{X} = P_X X, \bar{y} = P_y y$, prior to running Lasso. Thus, the general strategy of these methods is

$$\text{Preconditioned Lasso:} \quad \hat{\bar{\beta}} = \text{argmin}_{\beta \in \mathbb{R}^p} \frac{1}{2n} \left\| \bar{y} - \bar{X}\beta \right\|_2^2 + \bar{\lambda} \left\| \beta \right\|_1 . \quad (4)$$

Although this class of algorithms often compares favorably to the Lasso in practice, our theoretical understanding of them is at present still fairly poor. Huang and Jojic [5], for example, consider only empirical evaluations, while both Jia and Rohe [6] and Paul et al. [9] consider asymptotic consistency under various assumptions. Important and necessary as they are, consistency results do not provide insight into the relative performance of Preconditioned Lasso variants for finite data sets. In this paper we provide a new theoretical basis for making such comparisons. Although the focus of the paper is on problems of the form of Eq. (4), we note that the core ideas can also be applied to algorithms that right-multiply $X$ and/or $y$ with some matrices (e.g., [4, 11]).

For particular instances of $X, \beta^*$, we want to discover whether a given Preconditioned Lasso algorithm following Eq. (4) improves or degrades signed support recovery relative to the standard Lasso of Eq. (2). A major roadblock to a one-to-one comparison are the auxiliary penalty parameters, $\lambda, \bar{\lambda}$, which trade off the $\ell_1$ penalty to the quadratic objective in both Eq. (2) and Eq. (4). A correct choice of penalty parameter is essential for signed support recovery: If it is too small, the algorithm behaves like ordinary least squares; if it is too large, the estimated support may be empty. Unfortunately, in all but the simplest cases, pre-multiplying data $X, y$ by matrices $P_X, P_y$ changes the relative geometry of the $\ell_1$ penalty contours to the elliptical objective contours in a nontrivial way. Suppose we wanted to compare the Lasso to the Preconditioned Lasso by choosing for each $\lambda$ in Eq. (2) a suitable, matching $\bar{\lambda}$ in Eq. (4). For a *fair* comparison, the resulting mapping would have to capture the change of relative geometry induced by preconditioning of $X, y$, i.e. $\bar{\lambda} = f(\lambda, X, y, P_X, P_y)$. It seems difficult to theoretically characterize such a mapping. Furthermore, it seems unlikely that a comparative framework could be built by independently choosing "ideal" penalty parameters $\lambda, \bar{\lambda}$: Meinshausen and Bühlmann [8], for example, demonstrate that a seemingly reasonable oracle estimator of $\lambda$ will not lead to consistent support recovery in the Lasso. In the Preconditioned Lasso literature this problem is commonly sidestepped either by resorting to asymptotic comparisons [6, 9], empirically comparing regularization paths [5], or using model-selection techniques which aim to choose reasonably "good" matching penalty parameters [6]. We deem these approaches to be unsatisfactory—asymptotic and empirical analyses provide limited insight, and model selection strategies add a layer of complexity that may lead to unfair comparisons.

It is our view that all of these approaches place unnecessary emphasis on particular choices of penalty parameter. In this paper we propose an alternative strategy that instead compares the Lasso to the Preconditioned Lasso by comparing *data-dependent upper and lower penalty parameter bounds*. Specifically, we give bounds $(\lambda_u, \lambda_l)$ on $\lambda$ so that the Lasso in Eq. (2) is guaranteed to recover the signed support iff $\lambda_l < \lambda < \lambda_u$. Consequently, if $\lambda_l > \lambda_u$ signed support recovery is not possible. The Preconditioned Lasso in Eq. (4) uses data $\bar{X} = P_X X, \bar{y} = P_y y$ and will thus induce new bounds $(\bar{\lambda}_u, \bar{\lambda}_l)$ on $\bar{\lambda}$. The comparison of Lasso and Preconditioned Lasso on an instance $X, \beta^*$ then proceeds by suitably comparing the bounds on $\lambda$ and $\bar{\lambda}$. The advantage of this approach is that the upper and lower bounds are easy to compute, even though a general mapping between specific penalty parameters cannot be readily derived.

To demonstrate the effectiveness of our framework, we use it to analyze three Preconditioned Lasso algorithms [5, 6, 9]. Using our framework we make several contributions: (1) We confirm intuitions about advantages and disadvantages of the algorithms proposed in [5, 9]; (2) We show that for an SVD-based construction of $n \times p$ matrices $X$, the algorithm in [6] changes the bounds deterministically; (3) We show that in the context of our framework, this SVD-based construction can be thought of as a limit point of a Gaussian construction.

The paper is organized as follows. In Section 2 we will discuss three recent instances of Eq. (4). We outline our comparative framework in Section 3 and highlight some immediate consequences for [5] and [9] on general matrices $X$ in Section 4. More detailed comparisons can be made by considering a generative model for $X$. In Section 5 we introduce such a model based on a block-wise SVD of $X$ and then analyze [6] for specific instances of this generative model. Finally, we show that in terms of signed support recovery, this generative model can be thought of as a limit point of a Gaussian

construction. Section 6 concludes with some final thoughts. The proofs of all lemmas and theorems are in the supplementary material.

## 2 Preconditioned Lasso Algorithms

Our interest lies in the class of Preconditioned Lasso algorithms that is summarized by Eq. (4). Extensions to related algorithms, such as [4, 11] will follow readily. In this section we focus on three recent Preconditioned Lasso examples and instantiate the matrices $P_X, P_y$ appropriately. Detailed derivations can be found in the supplementary material. For later reference, we will denote each algorithm by the author initials.

**Huang and Jojic [5] (HJ).** Huang and Jojic proposed Correlation Sifting [5], which, although not presented as a preconditioning algorithm, can be rewritten as one. Let the SVD of $X$ be $X = UDV^\top$. Given an algorithm parameter $q$, let $U_\mathcal{A}$ be the set of $q$ smallest left singular vectors of $X$[1]. Then HJ amounts to setting

$$P_X = P_y = U_\mathcal{A} U_\mathcal{A}^\top. \tag{5}$$

**Paul et al. [9] (PBHT).** An earlier instance of the preconditioning idea was put forward by Paul et al. [9]. For some algorithm parameter $q$, let $\mathcal{A}$ be the $q$ column indices of $X$ with largest absolute correlation to $y$, (i.e., where $|X_j^\top y|/\|X_j\|_2$ is largest). Define $U_\mathcal{A}$ to be the $q$ largest left singular vectors of $X_\mathcal{A}$. With this, PBHT can be expressed as setting

$$P_X = I_{n\times n} \qquad P_y = U_\mathcal{A} U_\mathcal{A}^\top. \tag{6}$$

**Jia and Rohe [6] (JR).** Jia and Rohe [6] propose a preconditioning method that amounts to whitening the matrix $X$. If $X = UDV^\top$ is full rank, then JR defines[2]

$$P_X = P_y = U \left(DD^\top\right)^{-1/2} U^\top. \tag{7}$$

If $n < p$ then $\bar{X}\bar{X}^\top = P_X X X^\top P_X^\top \propto I_{n\times n}$ and if $n > p$ then $\bar{X}^\top \bar{X} = X^\top P_X^\top P_X X \propto I_{p\times p}$.

Both HJ and PBHT estimate a basis $U_\mathcal{A}$ for a $q$-dimensional subspace onto which they project $y$ and/or $X$. However, since the methods differ substantially in their assumptions, the estimators differ also. Empirical results in [5] and [9] suggest that the respective assumptions are useful in a variety of situations. In contrast, JR *reweights* the column space directions $U$ and requires no extra parameter $q$ to be estimated.

## 3 Comparative Framework

In this section we propose a new comparative approach for Preconditioned Lasso algorithms which avoids choosing particular penalty parameters $\lambda, \bar{\lambda}$. We first derive upper and lower bounds for $\lambda$ and $\bar{\lambda}$ respectively so that signed support recovery can be guaranteed iff $\lambda$ and $\bar{\lambda}$ satisfy the bounds. We then compare estimators by comparing the resulting bounds.

### 3.1 Conditions for signed support recovery

Before proceeding, we make some definitions motivated by Wainwright [12]. Suppose that the support set of $\beta^*$ is $S \triangleq S(\beta^*)$, with $|S| = k$. To simplify notation, we will assume throughout that $S = \{1, \ldots, k\}$ so that the corresponding off-support set is $S^c = \{1, \ldots, p\} \setminus S$, with $|S^c| = p - k$. Denote by $X_j$ column $j$ of $X$ and by $X_\mathcal{A}$ the submatrix of $X$ consisting of columns indexed by set $\mathcal{A}$. Define the following variables: For all $j \in S^c$ and $i \in S$, let

$$\mu_j = X_j^\top X_S (X_S^\top X_S)^{-1} \text{sgn}(\beta_S^*) \qquad \eta_j = X_j^\top \left(I_{n\times n} - X_S(X_S^\top X_S)^{-1} X_S^\top\right) \frac{w}{n} \tag{8}$$

$$\gamma_i = e_i^\top \left(\frac{1}{n} X_S^\top X_S\right)^{-1} \text{sgn}(\beta_S^*) \qquad \epsilon_i = e_i^\top \left(\frac{1}{n} X_S^\top X_S\right)^{-1} X_S^\top \frac{w}{n}. \tag{9}$$

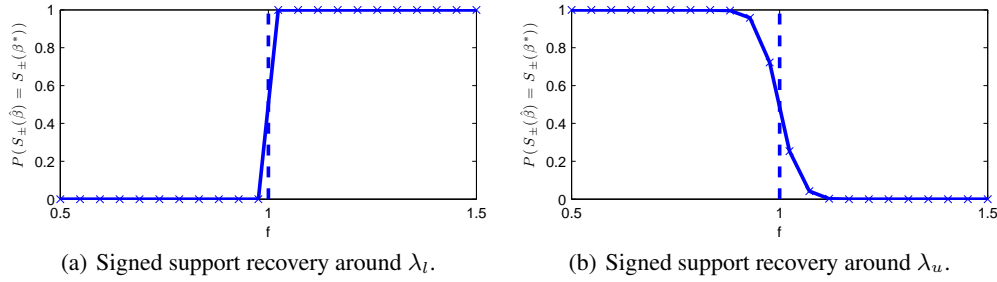

(a) Signed support recovery around $\lambda_l$.                    (b) Signed support recovery around $\lambda_u$.

Figure 1: Empirical evaluation of the penalty parameter bounds of Lemma 1. For each of 500 synthetic Lasso problems ($n = 300, p = 1000, k = 10$) we computed $\lambda_l, \lambda_u$ as per Lemma 1. Then we ran Lasso using penalty parameters $f\lambda_l$ in Figure (a) and $f\lambda_u$ in Figure (b), where the factor $f = 0.5, \ldots, 1.5$. The figures show the empirical probability of signed support recovery as a function of the factor $f$ for both $\lambda_l$ and $\lambda_u$. As expected, the probabilities change sharply at $f = 1$.

For the traditional Lasso of Eq. (2), results in (for example) Wainwright [12] connect settings of $\lambda$ with instances of $X, \beta^*, w$ to certify whether or not Lasso will recover the signed support. We invert these results and, for particular instances of $X, \beta^*, w$, derive bounds on $\lambda$ so that signed support recovery is guaranteed if and only if the bounds are satisfied. Specifically, we prove the following Lemma in the supplementary material.

**Lemma 1.** *Suppose that $X_S^\top X_S$ is invertible, $|\mu_j| < 1, \forall j \in S^c$, and $sgn(\beta_i^*)\gamma_i > 0, \forall i \in S$. Then the Lasso has a unique solution $\hat{\beta}$ which recovers the signed support (i.e., $S_\pm(\hat{\beta}) = S_\pm(\beta^*)$) if and only if $\lambda_l < \lambda < \lambda_u$, where*

$$\lambda_l = \max_{j \in S^c} \frac{\eta_j}{(2[\![\eta_j > 0]\!] - 1) - \mu_j} \qquad \lambda_u = \min_{i \in S} \left| \frac{\beta_i^* + \epsilon_i}{\gamma_i} \right|_+, \tag{10}$$

$[\![\cdot]\!]$ *denotes the indicator function and $|\cdot|_+ = \max(0, \cdot)$ denotes the hinge function. On the other hand, if $X_S^\top X_S$ is not invertible, then the signed support cannot in general be recovered.*

Lemma 1 recapitulates well-worn intuitions about when the Lasso has difficulty recovering the signed support. For instance, assuming that $w$ has symmetric distribution with mean 0, if $1 - |\mu_j|$ is small (i.e., the irrepresentable condition almost fails to hold), then $\lambda_l$ will tend to be large. In extreme cases we might have $\lambda_l > \lambda_u$ so that signed support recovery is impossible. Figure 1 empirically validates the bounds of Lemma 1 by estimating probabilities of signed support recovery for a range of penalty parameters on synthetic Lasso problems.

### 3.2 Comparisons

In this paper we propose to compare a preconditioning algorithm to the traditional Lasso by comparing the penalty parameter bounds produced by Lemma 1. As highlighted in Eq. 4, the preconditioning framework runs Lasso on modified variables $\bar{X} = P_X X, \bar{y} = P_y y$. For the purpose of applying Lemma 1, these transformations induce a new noise vector

$$\bar{w} = \bar{y} - \bar{X}\beta^* = P_y (X\beta^* + w) - P_X X \beta^*. \tag{11}$$

Note that if $P_X = P_y$ then $\bar{w} = P_y w$. Provided the conditions of Lemma 1 hold for $\bar{X}, \beta^*$ we can define updated variables $\bar{\mu}_j, \bar{\gamma}_i, \bar{\eta}_j, \bar{\epsilon}_i$ from which the bounds $\bar{\lambda}_u, \bar{\lambda}_l$ on the penalty parameter $\bar{\lambda}$ can be derived. In order for our comparison to be scale-invariant, we will compare algorithms by *ratios* of resulting penalty parameter bounds. That is, we deem a Preconditioned Lasso algorithm to be more effective than the traditional Lasso if $\bar{\lambda}_u/\bar{\lambda}_l > \lambda_u/\lambda_l$. Intuitively, the upper bound $\bar{\lambda}_u$ is then *disproportionately* larger than $\bar{\lambda}_l$ relative to $\lambda_u$ and $\lambda_l$, which in principle allows easier tuning of $\bar{\lambda}$[3]. We will later encounter the special case $\bar{\lambda}_u \neq 0, \bar{\lambda}_l = 0$ in which case we define $\bar{\lambda}_u/\bar{\lambda}_l \triangleq \infty$ to indicate that the preconditioned problem is very easy. If $\bar{\lambda}_u/\bar{\lambda}_l < 1$ then signed support recovery is in general impossible. Finally, to match this intuition, we define $\bar{\lambda}_u/\bar{\lambda}_l \triangleq 0$ if $\bar{\lambda}_u = \bar{\lambda}_l = 0$.

## 4 General Comparisons

We begin our comparisons with some immediate consequences of Lemma 1 for HJ and PBHT. In order to highlight the utility of the proposed framework, we focus in this section on special cases of $P_X, P_y$. The framework can of course also be applied to general matrices $P_X, P_y$. As we will see, both HJ and PBHT have the potential to improve signed support recovery relative to the traditional Lasso, provided the matrices $P_X, P_y$ are suitably estimated. The following notation will be used during our comparisons: We will write $\bar{A} \preceq A$ to indicate that random variable $A$ stochastically dominates $\bar{A}$, that is, $\forall t \ \ \mathbb{P}(\bar{A} \geq t) \leq \mathbb{P}(A \geq t)$. We also let $U_S$ be a minimal basis for the column space of the submatrix $X_S$, and define $\text{span}(U_S) = \{x \mid \exists c \in \mathbb{R}^k \text{ s.t. } x = U_S c\} \subseteq \mathbb{R}^n$. Finally, we let $U_{S^c}$ be a minimal basis for the orthogonal complement of $\text{span}(U_S)$.

**Consequences for HJ.** Recall from Section 2 that HJ uses $P_X = P_y = U_\mathcal{A} U_\mathcal{A}^\top$, where $U_\mathcal{A}$ is a column basis estimated from $X$. We have the following theorem:

**Theorem 1.** *Suppose that the conditions of Lemma 1 are met for a fixed instance of $X, \beta^*$. If $\text{span}(U_S) \subseteq \text{span}(U_\mathcal{A})$, then after preconditioning using HJ the conditions continue to hold, and*

$$\frac{\lambda_u}{\lambda_l} \preceq \frac{\bar{\lambda}_u}{\bar{\lambda}_l}, \tag{12}$$

*where the stochasticity on both sides is due to independent noise vectors $w$. On the other hand, if $X_S^\top P_X^\top P_X X_S$ is not invertible, then HJ cannot in general recover the signed support.*

We briefly sketch the proof of Theorem 1. If $\text{span}(U_S) \subseteq \text{span}(U_\mathcal{A})$ then plugging in the definition of $P_X$ into $\bar{\mu}_j, \bar{\gamma}_i, \bar{\eta}_j, \bar{\epsilon}_i$, one can derive the following

$$\bar{\mu}_j = \mu_j \qquad\qquad \bar{\gamma}_i = \gamma_i \tag{13}$$

$$\bar{\eta}_j = X_j^\top \left(I_{n \times n} - U_S U_S^\top\right) U_\mathcal{A} U_\mathcal{A}^\top \frac{w}{n} \qquad \bar{\epsilon}_i = \epsilon_i. \tag{14}$$

If $\text{span}(U_\mathcal{A}) = \text{span}(U_S)$, then it is easy to see that $\bar{\eta}_j = 0$. Notice that because $\bar{\mu}_j$ and $\bar{\gamma}_i$ are unchanged, if the conditions of Lemma 1 hold for the original Lasso problem (i.e., $X_S^\top X_S$ is invertible, $|\mu_j| < 1 \ \ \forall j \in S^c$ and $\text{sgn}(\beta_i^*)\gamma_i > 0 \ \ \forall i \in S$), they will continue to hold for the preconditioned problem. Suppose then that the conditions set forth in Lemma 1 are met. With some additional work one can show that

$$\bar{\lambda}_u = \min_{i \in S} \left| \frac{\beta_i^* + \bar{\epsilon}_i}{\bar{\gamma}_i} \right|_+ = \lambda_u \qquad\qquad \bar{\lambda}_l = \max_{j \in S^c} \frac{\bar{\eta}_j}{(2[\![\bar{\eta}_j > 0]\!] - 1) - \bar{\mu}_j} \preceq \lambda_l. \tag{15}$$

The result then follows by showing that $\bar{\lambda}_l, \lambda_l$ are both independent of $\bar{\lambda}_u = \lambda_u$. Note that if $\text{span}(U_\mathcal{A}) = \text{span}(U_S)$, then $\bar{\lambda}_l = 0$ and so $\bar{\lambda}_u / \bar{\lambda}_l \triangleq \infty$. In the more common case when $\text{span}(U_S) \not\subseteq \text{span}(U_\mathcal{A})$ the performance of the Lasso depends on how misaligned $U_\mathcal{A}$ and $U_S$ are. In extreme cases, $X_S^\top P_X^\top P_X X_S$ is singular and so signed support recovery is not in general possible.

**Consequences for PBHT.** Recall from Section 2 that PBHT uses $P_X = I_{n \times n}, P_y = U_\mathcal{A} U_\mathcal{A}^\top$, where $U_\mathcal{A}$ is a column basis estimated from $X$. We have the following theorem.

**Theorem 2.** *Suppose that the conditions of Lemma 1 are met for a fixed instance of $X, \beta^*$. If $\text{span}(U_S) \subseteq \text{span}(U_\mathcal{A})$, after preconditioning using PBHT the conditions continue to hold, and*

$$\frac{\lambda_u}{\lambda_l} \preceq \frac{\bar{\lambda}_u}{\bar{\lambda}_l}, \tag{16}$$

*where the stochasticity on both sides is due to independent noise vectors $w$. On the other hand, if $\text{span}(U_{S^c}) = \text{span}(U_\mathcal{A})$, then PBHT cannot recover the signed support.*

As before, we sketch the proof to build some intuition. Because PBHT does not set $P_X = P_y$ as HJ does, there is no danger of $X_S^\top P_X^\top P_X X_S$ becoming singular. On the other hand, this complicates the form of the induced noise vector $\bar{w}$. Plugging $P_X$ and $P_y$ into Eq. (11), we find $\bar{w} = (U_\mathcal{A} U_\mathcal{A}^\top - I_{n \times n}) X \beta^* + U_\mathcal{A} U_\mathcal{A}^\top w$. However, even though the noise has a more complicated form, derivations in the supplementary material show that if $\text{span}(U_S) \subseteq \text{span}(U_\mathcal{A})$, then

$$\bar{\mu}_j = \mu_j \qquad\qquad \bar{\gamma}_i = \gamma_i \tag{17}$$

$$\bar{\eta}_j = X_j^\top \left(I_{n \times n} - U_S U_S^\top\right) U_\mathcal{A} U_\mathcal{A}^\top \frac{w}{n} \qquad \bar{\epsilon}_i = \epsilon_i. \tag{18}$$

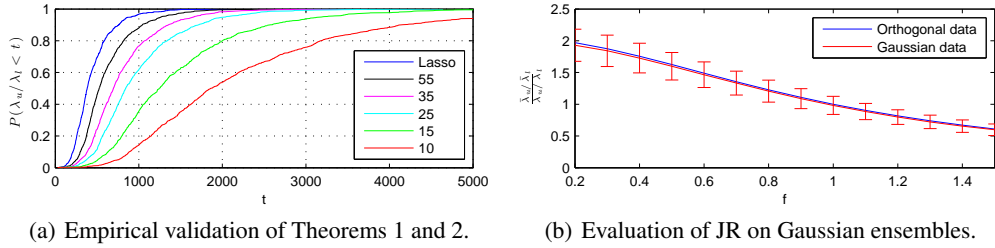

(a) Empirical validation of Theorems 1 and 2.    (b) Evaluation of JR on Gaussian ensembles.

Figure 2: Experimental evaluations. Figure (a) shows empirical c.d.f.'s of penalty parameter bounds ratios estimated from 1000 variable selection problems. Each problem consists of Gaussians $X$ and $w$, and $\beta^*$, with $n = 100, p = 300, k = 5$. The blue curve shows the c.d.f. for $\lambda_u/\lambda_l$ estimated on the original data (Lasso). Then we projected the data using $P_X = P_y = U_{\mathcal{A}} U_{\mathcal{A}}^\top$, where $\mathrm{span}(U_S) \subseteq \mathrm{span}(U_{\mathcal{A}})$ but $\dim(U_{\mathcal{A}}) = \dim(\mathrm{span}(U_{\mathcal{A}}))$ is variable (see legend), and estimated the resulting c.d.f. for the updated bounds ratio $\bar{\lambda}_u/\bar{\lambda}_l$. As predicted by Theorems 1 and 2, $\lambda_u/\lambda_l \preceq \bar{\lambda}_u/\bar{\lambda}_l$. In Figure (b) the blue curve shows the scale factor $(p-k)/(n + p\kappa^2 - k)$ predicted by Theorem 3 for problems constructed from Eq. (19) for $\kappa = f\sqrt{1 - (n/p)}$. The red curve plots the corresponding factor estimated from the Gaussian construction in Eq. (25) ($n = 100, m = 2000, p = 200, k = 5$) using the same $\Sigma_S, \Sigma_{S^c}$ as in Theorem 3, averaged over 50 problem instances and with error bars for one standard deviation. As in Theorem 3, the factor is approximately 1 if $f = 1$.

As with HJ, if $\mathrm{span}(U_{\mathcal{A}}) = \mathrm{span}(U_S)$, then $\bar{\eta}_j = 0$. Because $\bar{\mu}_j$ and $\bar{\gamma}_i$ are again unchanged, the conditions of Lemma 1 will continue to hold for the preconditioned problem if they hold for the original Lasso problem. With the previous equalities established, the remainder of the proof is identical to that of Theorem 1. The fact that the above $\bar{\mu}_j, \bar{\eta}_j, \bar{\gamma}_i, \bar{\epsilon}_i$ are identical to those of HJ depends crucially on the fact that $\mathrm{span}(U_S) \subseteq \mathrm{span}(U_{\mathcal{A}})$. In general the values will differ because PBHT sets $P_X = I_{n \times n}$, but HJ does not.

On the other hand, if $\mathrm{span}(U_S) \not\subseteq \mathrm{span}(U_{\mathcal{A}})$ then the distribution of $\bar{\epsilon}_i$ depends on how misaligned $U_{\mathcal{A}}$ and $U_S$ are. In the extreme case when $\mathrm{span}(U_{S^c}) = \mathrm{span}(U_{\mathcal{A}})$, one can show that $\bar{\epsilon}_i = -\beta_i^*$, which results in $\bar{\lambda}_u = 0, \bar{\lambda}_l \preceq \lambda_l$. Because $\mathbb{P}(\bar{\lambda}_l \geq 0) = 1$, signed support recovery is not possible.

**Remarks.** Our theoretical analyses show that both HJ and PBHT can indeed lead to improved signed support recovery relative to the Lasso on finite datasets. To underline our findings, we empirically validate Theorems 1 and 2 in Figure 2(a), where we plot estimated c.d.f.'s for penalty parameter bounds ratios of Lasso and Preconditioned Lasso for various subspaces $U_{\mathcal{A}}$. Our theorems focussed on specific settings of $P_X, P_y$ and ignored others. In general, the gains of HJ and PBHT over Lasso depend on how much the decoy signals in $X_{S^c}$ are suppressed and how much of the true signal due to $X_S$ is preserved. Further comparison of HJ and PBHT must thus analyze how the subspaces $\mathrm{span}(U_{\mathcal{A}})$ are estimated in the context of the assumptions made in [5] and [9]. A final note concerns the dimension of the subspace $\mathrm{span}(U_{\mathcal{A}})$. Both HJ and PBHT were proposed with the implicit goal of finding a basis $U_{\mathcal{A}}$ that has the same span as $U_S$. This of course requires estimating $|S| = k$ by $q$, which adds another layer of complexity to these algorithms. Theorems 1 and 2 suggest that underestimating $k$ can be more detrimental to signed support recovery than overestimating it. By overestimating $q > k$, we can trade off milder improvement when $\mathrm{span}(U_S) \subseteq \mathrm{span}(U_{\mathcal{A}})$ against poor behavior should we have $\mathrm{span}(U_S) \not\subseteq \mathrm{span}(U_{\mathcal{A}})$.

## 5    Model-Based Comparisons

In the previous section we used Lemma 1 in conjunction with assumptions on $U_{\mathcal{A}}$ to make statements about HJ and PBHT. Of course, the quality of the estimated $U_{\mathcal{A}}$ depends on the specific instances $X, \beta^*, w$, which hinders a general analysis. Similarly, a direct application of Lemma 1 to JR yields bounds that exhibit strong $X$ dependence. It is possible to crystallize prototypical examples by specializing $X$ and $w$ to come from a generative model. In this section we briefly present this model and will show the resulting penalty parameter bounds for JR.

## 5.1 Generative model for $X$

As discussed in Section 2, many preconditioning algorithms can be phrased as truncating or reweighting column subspaces associated with $X$ [5, 6, 9]. This suggests that a natural generative model for $X$ can be formulated in terms of the SVD of submatrices of $X$.

Assume $p - k > n$ and let $\Sigma_S, \Sigma_{S^c}$ be fixed-spectrum matrices of dimension $n \times k$ and $n \times p - k$ respectively. We will assume throughout this paper that the top left "diagonal" entries of $\Sigma_S, \Sigma_{S^c}$ are positive and the remainder is zero. Furthermore, we let $U, V_S, V_{S^c}$ be orthonormal bases of dimension $n \times n, k \times k$ and $p - k \times p - k$ respectively. We assume that these bases are chosen uniformly at random from the corresponding Stiefel manifold. As before and without loss of generality, suppose $S = \{1, \ldots, k\}$. Then we let the Lasso problem be

$$y = X\beta^* + w \quad \text{with} \quad X = U\left[\Sigma_S V_S^\top, \Sigma_{S^c} V_{S^c}^\top\right] \quad w \sim \mathcal{N}(0, \sigma^2 I_{n \times n}), \tag{19}$$

To ensure that the column norms of $X$ are controlled, we compute the spectra $\Sigma_S, \Sigma_{S^c}$ by normalizing spectra $\hat{\Sigma}_S$ and $\hat{\Sigma}_{S^c}$ with arbitrary positive elements on the diagonal. Specifically, we let

$$\Sigma_S = \frac{\hat{\Sigma}_S}{\|\hat{\Sigma}_S\|_F}\sqrt{kn} \qquad \Sigma_{S^c} = \frac{\hat{\Sigma}_{S^c}}{\|\hat{\Sigma}_{S^c}\|_F}\sqrt{(p-k)n}. \tag{20}$$

We verify in the supplementary material that with these assumptions the squared column norms of $X$ are in expectation $n$ (provided the orthonormal bases are chosen uniformly at random).

**Intuition.** Note that any matrix $X$ can be decomposed using a block-wise SVD as

$$X = [X_S, X_{S^c}] = U\left[\Sigma_S V_S^\top, T\Sigma_{S^c} V_{S^c}^\top\right], \tag{21}$$

with orthonormal bases $U, T, V_S, V_{S^c}$. Our model in Eq. (19) is only a minor restriction of this model, where we set $T = I_{n \times n}$. To develop more intuition, let us temporarily set $V_S = I_{k \times k}$, $V_{S^c} = I_{p-k \times p-k}$. Then $X = [X_S, X_{S^c}] = U[\Sigma_S, \Sigma_{S^c}]$ and we see that up to scaling $X_S$ equals the first $k$ columns of $X_{S^c}$. The difficulty for Lasso thus lies in correctly selecting the columns in $X_S$, which are highly correlated with the first few columns in $X_{S^c}$.

## 5.2 Piecewise constant spectra

For notational clarity we will now focus on a special case of the above model. To begin, we develop some notation. In previous sections we used $U_S$ to denote a basis for the column space of $X_S$. We will continue to use this notation, and let $U_S$ contain the first $k$ columns of $U$. Accordingly, we denote the last $n - k$ columns of $U$ by $U_{S^c}$. We let the diagonal elements of $\Sigma_S, \hat{\Sigma}_S, \Sigma_{S^c}, \hat{\Sigma}_{S^c}$ be identified by their column indices. That is, the diagonal entries $\sigma_{S,c}$ of $\Sigma_S$ and $\hat{\sigma}_{S,c}$ of $\hat{\Sigma}_S$ are indexed by $c \in \{1, \ldots, k\}$; the diagonal entries $\sigma_{S^c,c}$ of $\Sigma_{S^c}$ and $\hat{\sigma}_{S^c,c}$ of $\hat{\Sigma}_{S^c}$ are indexed by $c \in \{1, \ldots, n\}$. Each of the diagonal entries in $\Sigma_S, \Sigma_{S^c}$ is associated with a column of $U$. The set of diagonal entries of $\Sigma_S$ and $\Sigma_{S^c}$ associated with $U_S$ is $\sigma(S) = \{1, \ldots, k\}$ and the set of diagonal entries in $\Sigma_{S^c}$ associated with $U_{S^c}$ is $\sigma(S^c) = \{1, \ldots, n\} \backslash \sigma(S)$. We will construct spectrum matrices $\Sigma_S, \Sigma_{S^c}$ that are piecewise constant on their diagonals. For some $\kappa \geq 0$, we let $\hat{\sigma}_{S,i} = 1, \hat{\sigma}_{S^c,i} = \kappa \; \forall i \in \sigma(S)$ and $\hat{\sigma}_{S^c,j} = 1 \; \forall j \in \sigma(S^c)$.

**Consequences for JR.** Recall that for JR, if $X = UDV^\top$, then $P_X = P_y = U\left(DD^\top\right)^{-1/2}U^\top$. We have the following theorem.

**Theorem 3.** *Assume the Lasso problem was generated according to the generative model of Eq. (19) with $\forall i \in \sigma(S)$, $\hat{\sigma}_{S,i} = 1$, $\hat{\sigma}_{S^c,i} = \kappa$ and $\forall j \in \sigma(S^c)$, $\hat{\sigma}_{S^c,j} = 1$ and that $\kappa < \sqrt{n-k}/\sqrt{k(p-k-1)}$. Then the conditions of Lemma 1 hold before and after preconditioning using JR. Moreover,*

$$\frac{\bar{\lambda}_u}{\bar{\lambda}_l} = \frac{(p-k)}{n + p\kappa^2 - k}\frac{\lambda_u}{\lambda_l}. \tag{22}$$

In other words, JR deterministically scales the ratio of penalty parameter bounds. The proof idea is as follows. It is easy to see that $X_S^\top X_S$ is always invertible. Furthermore, one can show that if

$\kappa < \sqrt{n-k}/\sqrt{k(p-k-1)}$, we have $|\mu_j| < 1, \forall j \in S^c$ and $\operatorname{sgn}(\beta_i^*)\gamma_i > 0, \forall i \in S$. Thus, by our assumptions, the preconditions of Lemma 1 are satisfied for the original Lasso problem. Plugging in the definitions of $\Sigma_S, \Sigma_{S^c}$ into Eq. (19) we find that the SVD becomes $X = UDV^\top$, where $U$ is the same column basis as in Eq. (19), and the diagonal elements of $D$ are determined by $\kappa$. Substituting this into the definitions of $\bar{\mu}_j, \bar{\gamma}_i, \bar{\eta}_j, \bar{\epsilon}_i$, we have that after preconditioning using JR

$$\bar{\mu}_j = \mu_j \qquad\qquad \bar{\gamma}_i = \left( n + \frac{n(p-k)\kappa^2}{k\kappa^2 + n - k} \right) \gamma_i \qquad (23)$$

$$\bar{\eta}_j = \frac{(k\kappa^2 + n - k)}{n(p-k)}\eta_j \qquad \bar{\epsilon}_i = \epsilon_i. \qquad (24)$$

Thus, if the conditions of Lemma 1 hold for $X, \beta^*$, they will continue to hold after preconditioning using JR. Furthermore, notice that $(2[\![\bar{\eta}_j > 0]\!] - 1) - \bar{\mu}_j = (2[\![\eta_j > 0]\!] - 1) - \mu_j$. Applying Lemma 1 then gives the new ratio $\bar{\lambda}_u/\bar{\lambda}_l$ as claimed. According to Theorem 3 the ratio $\bar{\lambda}_u/\bar{\lambda}_l$ will be larger than $\lambda_u/\lambda_l$ iff $\kappa < \sqrt{1 - (n/p)}$. Indeed, if $\kappa = \sqrt{1 - (n/p)}$ then $P_X = P_y \propto I_{n \times n}$ and so JR coincides with standard Lasso.

### 5.3 Extension to Gaussian ensembles

The construction in Eq. (19) uses an orthonormal matrix $U$ as the column basis of $X$. At first sight this may appear to be restrictive. However, as we show in the supplementary material, one can construct Lasso problems using a Gaussian basis $W^m$ which lead to penalty parameter bounds ratios that converge in distribution to those of the Lasso problem in Eq. (19). For some fixed $\beta^*$, $V_S$, $V_{S^c}$, $\Sigma_S$ and $\Sigma_{S^c}$, generate two independent problems: One using Eq. (19), and one according to

$$y^m = X^m \beta^* + w^m \text{ with } X^m = \frac{1}{\sqrt{n}} W^m \left[ \Sigma_S V_S^\top, \Sigma_{S^c} V_{S^c}^\top \right] \qquad w^m \sim \mathcal{N}\left( 0, \sigma^2 \frac{m}{n} I_{m \times m} \right), \quad (25)$$

where $W^m$ is an $m \times n$ standard Gaussian ensemble. Note that an $X$ so constructed is low rank if $n < p$. The latter generative model bears some resemblance to Gaussian models considered in Paul et al. [9] (Eq. (7)) and Jia and Rohe [6] (Proposition 2). Note that while the problem in Eq. (19) uses $n$ observations with noise variance $\sigma^2$, Eq. (25) has $m$ observations with noise variance $\sigma^2 m/n$. The increased variance is necessary because the matrix $W^m$ has expected column length $m$, while columns in $U$ are of length 1. We will think of $n$ as fixed and will let $m \to \infty$. Let the penalty parameter bounds ratio induced by the problem in Eq. (19) be $\lambda_u/\lambda_l$ and that induced by Eq. (25) be $\lambda_u^m/\lambda_l^m$. Then we have the following result.

**Theorem 4.** *Let $V_S$, $V_{S^c}$, $\Sigma_S$, $\Sigma_{S^c}$ and $\beta^*$ be fixed. If the conditions of Lemma 1 hold for $X, \beta^*$, then for $m$ large enough they will hold for $X^m, \beta^*$. Furthermore, as $m \to \infty$*

$$\frac{\lambda_u^m}{\lambda_l^m} \xrightarrow{d} \frac{\lambda_u}{\lambda_l}, \qquad (26)$$

*where the stochasticity on the left is due to $W^m, w^m$ and on the right is due to $w$.*

Thus, with respect to the bounds ratio $\lambda_u/\lambda_l$, the construction of Eq. (19) can be thought of as the limiting construction of Gaussian Lasso problems in Eq. (25) for large $m$. As such, we believe that Eq. (19) is a useful proxy for less restrictive generative models. Indeed, as the experiment in Figure 2(b) shows, Theorem 3 can be used to predict the scaling factor for penalty parameter bounds ratios (i.e., $\left( \bar{\lambda}_u/\bar{\lambda}_l \right) / (\lambda_u/\lambda_l)$) with good accuracy even for Gaussian ensembles.

## 6 Conclusions

This paper proposes a new framework for comparing Preconditioned Lasso algorithms to the standard Lasso which skirts the difficulty of choosing penalty parameters. By eliminating this parameter from consideration, finite data comparisons can be greatly simplified, avoiding the use of model selection strategies. To demonstrate the framework's usefulness, we applied it to a number of Preconditioned Lasso algorithms and in the process confirmed intuitions and revealed fragilities and mitigation strategies. Additionally, we presented an SVD-based generative model for Lasso problems that can be thought of as the limit point of a less restrictive Gaussian model. We believe this work to be a first step towards a comprehensive theory for evaluating and comparing Lasso-style algorithms and believe that the strategy can be extended to comparing other penalized likelihood methods on finite datasets.

## Footnotes

[1]The choice of *smallest* singular vectors is considered for matrices $X$ with sharply decaying spectrum.

[2]We note that Jia and Rohe [6] let $D$ be square, so that it can be directly inverted. If $X$ is not full rank, the pseudo-inverse of $D$ can be used.

[3]Other functions of $\lambda_l, \lambda_u$ and $\bar{\lambda}_l, \bar{\lambda}_u$ could also be considered. However, we find the ratio to be a particularly intuitive measure.

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
