[Supplementary Material]

# A Comparative Framework for Preconditioned Lasso Algorithms
# — Supplementary Material —

## 1  Preconditioning Algorithms

In this section we briefly show how to express PBHT and HJ in a framework that runs Lasso on modified variables $P_X X$ and $P_y y$.

### 1.1  Huang and Jojic [1] (HJ)

Consider the SVD $X = UDV^\top$, where $U$ is $n \times n$, $V$ is $p \times p$ and $D$ is an $n \times p$ "diagonal" matrix with entries $d_1 < \ldots < d_n$[1]. Define two groups of left and right singular vectors associated with the $q$ smallest and $n - q$ largest singular values. Let the groups be defined by $U_q, U_{n-q}$ and $V_q, V_{n-q}$. Suppose HJ chooses as its row-basis the $n - q$ largest right singular vectors, $V_{n-q}$. Then, from Table 1 of Huang and Jojic [1] we find that

$$Z = XV_{n-q} = U_{n-q}\text{diag}(\{d_j\}_{j>q}) \tag{1}$$

$$\bar{X} = R = X - ZV_{n-q}^\top \tag{2}$$

$$= X - U_{n-q}\text{diag}(\{d_j\}_{j>q})V_{n-q}^\top \tag{3}$$

$$= U_q\text{diag}(\{d_i\}_{i\leq q})V_q^\top \tag{4}$$

$$= U_q U_q^\top X \tag{5}$$

$$\bar{y} = y - Z(Z^\top Z)^{-1}Z^\top y \tag{6}$$

$$= y - U_{n-q}U_{n-q}^\top y \tag{7}$$

$$= U_q U_q^\top y \tag{8}$$

So HJ sets $P_X = P_y = U_{\mathcal{A}}U_{\mathcal{A}}^\top$ for a suitably estimated subspace $U_{\mathcal{A}}$

### 1.2  Paul et al. [2] (PBHT)

Suppose PBHT identifies as $X_q$ the $q$ columns of $X$ that are most correlated with $y$ (i.e., where $|X_j^\top y|/\|X_j\|_2$ is largest). Consider the SVD $X_q = UDV^\top$, where $U$ is $n \times n$, $V$ is $q \times q$ and $D$ is $n \times q$. Paul et al. [2] use $V$ to find the projection matrix $P_q$. Let the columns of $V$ be denoted by $v_{q'}$ and those of $U$ by $u_{q'}$. From Section 4.5 and Eq. (13) in Paul el al. [2][2].

$$P_q = \sum_{q'=1}^{q} \frac{1}{\|X_q v_{q'}\|_2^2} X_q v_{q'} v_{q'}^\top X_q^\top \tag{9}$$

$$= \sum_{q'=1}^{q} \frac{1}{d_{q'}^2} u_{q'} d_{q'}^2 u_{q'}^\top = U_q U_q^\top \tag{10}$$

$$\bar{X} = X \tag{11}$$

$$\bar{y} = P_q y = U_q U_q^\top y, \tag{12}$$

where $U_q$ consists of the first $q$ columns of $U$. Thus, PBHT sets $P_X = I_{n\times n}$ and $P_y = U_{\mathcal{A}}U_{\mathcal{A}}^\top$ for a suitably estimated subspace $U_{\mathcal{A}}$

# 2 Proof of Lemma 1

**Lemma 1.** *Suppose that $X_S^\top X_S$ is invertible, $|\mu_j| < 1$ $\forall j \in S^c$ and $sgn(\beta_i^*)\gamma_i > 0$ $\forall i \in S$. Then the Lasso has a unique solution $\hat{\beta}$ which recovers the signed support (i.e., $S_\pm(\hat{\beta}) = S_\pm(\beta^*)$) if and only if $\lambda_l < \lambda < \lambda_u$, where*

$$\lambda_l = \max_{j \in S^c} \frac{\eta_j}{(2[\![\eta_j > 0]\!] - 1) - \mu_j} \qquad \lambda_u = \min_{i \in S} \left| \frac{\beta_i^* + \epsilon_i}{\gamma_i} \right|_+, \qquad (13)$$

*$[\![\cdot]\!]$ denotes the indicator function and $|\cdot|_+ = \max(0, \cdot)$ denotes the hinge function. On the other hand, if $X_S^\top X_S$ is not invertible, then the signed support cannot in general be recovered.*

*Proof.* For a particular choice of $\lambda$, and instances $X, \beta^*, w$, Lemmas 2 and 3 of Wainwright give conditions under which Lasso produces a unique $\hat{\beta}$ which recovers the signed support. If $X_S^\top X_S$ is invertible, then by Lemmas 2 and 3

$$S_\pm(\hat{\beta}) = S_\pm(\beta^*) \iff \forall j \in S^c \ |Z_j| < 1 \text{ and } \forall i \in S \ sgn(\beta_i^* + \Delta_i) = sgn(\beta_i^*), \qquad (14)$$

where

$$Z_j \triangleq \mu_j + \frac{1}{\lambda}\eta_j \qquad (15)$$

$$\mu_j = X_j^\top X_S(X_S^\top X_S)^{-1}sgn(\beta_S^*) \qquad (16)$$

$$\eta_j = X_j^\top \left( I_{n \times n} - X_S(X_S^\top X_S)^{-1}X_S^\top \right) \frac{w}{n} \qquad (17)$$

$$\Delta_i \triangleq \epsilon_i - \lambda\gamma_i \qquad (18)$$

$$\epsilon_i = e_i^\top \left( \frac{1}{n}X_S^\top X_S \right)^{-1} \frac{1}{n}X_S^\top w \qquad (19)$$

$$\gamma_i = e_i^\top \left( \frac{1}{n}X_S^\top X_S \right)^{-1} sgn(\beta_S^*) \qquad (20)$$

We can invert Lemmas 2 and 3 and derive from them conditions on $\lambda$ so that signed support recovery can be guaranteed.

**Ensure** $\forall j \in S^c, |Z_j| < 1$

For this to hold, we need $\forall j \in S^c$,

$$|Z_j| = \left| \mu_j + \frac{1}{\lambda}\eta_j \right| < 1. \qquad (21)$$

Since we assumed that $|\mu_j| < 1$ $\forall j \in S^c$, we have:

**Case 1a:** $\eta_j \geq 0$

We need for every $j \in S^c$

$$\mu_j + \frac{1}{\lambda}\eta_j < 1 \qquad (22)$$

$$\frac{1}{\lambda}\eta_j < 1 - \mu_j \qquad (23)$$

$$\lambda > \frac{\eta_j}{1 - \mu_j} \qquad (24)$$

**Case 1b:** $\eta_j \leq 0$

We need for every $j \in S^c$

$$\mu_j + \frac{1}{\lambda}\eta_j > -1 \tag{25}$$

$$\frac{1}{\lambda}\eta_j > -1 - \mu_j \tag{26}$$

$$\lambda > -\frac{\eta_j}{1+\mu_j} = \frac{\eta_j}{-1-\mu_j}. \tag{27}$$

Combining, we need

$$\lambda > \lambda_l = \max_{j \in S^c} \frac{\eta_j}{(2[\![\eta_j > 0]\!] - 1) - \mu_j} \geq 0. \tag{28}$$

**Ensure** $\forall i \in S, \mathbf{sgn}(\beta_i^* + \Delta_i) = \mathbf{sgn}(\beta_i^*)$

Since we assumed $\mathrm{sgn}(\beta_i^*)\gamma_i > 0 \ \forall i \in S$, we have in particular that $\gamma_i \neq 0$. Then

**Case 2a:** $\beta_i^* > 0$

Since $\mathrm{sgn}(\beta_i^*)\gamma_i > 0$, we have $\gamma_i > 0$. Then we need

$$\beta_i^* + \Delta_i = \beta_i^* + \epsilon_i - \lambda\gamma_i > 0 \tag{29}$$
$$\lambda\gamma_i < \beta_i^* + \epsilon_i \tag{30}$$
$$\lambda < \frac{\beta_i^* + \epsilon_i}{\gamma_i} \tag{31}$$

**Case 2b:** $\beta_i^* < 0$

Since $\mathrm{sgn}(\beta_i^*)\gamma_i > 0$, we have $\gamma_i < 0$. We need

$$\beta_i^* + \Delta_i = \beta_i^* + \epsilon_i - \lambda\gamma_i < 0 \tag{32}$$
$$\lambda\gamma_i > \beta_i^* + \epsilon_i \tag{33}$$
$$\lambda < \frac{\beta_i^* + \epsilon_i}{\gamma_i} \tag{34}$$

Hence, overall we need

$$\lambda < \min_{i \in S} \frac{\beta_i^* + \epsilon_i}{\gamma_i}. \tag{35}$$

Although the previous equation could be used to make a definition for $\lambda_u$, it will be beneficial later if $\lambda_u \geq 0$. Because $\lambda_l \geq 0$, signed support recovery will not be possible whenever $\min_{i \in S}(\beta_i^* + \epsilon_i)/\gamma_i \leq 0$. Thus, we will define

$$\lambda_u = \min_{i \in S} \left| \frac{\beta_i^* + \epsilon_i}{\gamma_i} \right|_+, \tag{36}$$

where $|\cdot|_+ = \max(0, \cdot)$ is the hinge function. Signed support recovery occurs iff $\lambda_l < \lambda < \lambda_u$. On the other hand, if $X_S^\top X_S$ is not invertible, the columns of $X_S$ are linearly dependent and so the signed support cannot be recovered in general. $\qquad\square$

# 3 Proofs of Section 4

To simplify the proofs of Section 4, we will make repeated use of the following lemma.

**Lemma 2.** *Suppose $U, V$ are orthonormal bases for subspaces lying in $\mathbb{R}^n$. That is, $U$ is $n \times q$, with $q \leq n$ and $U^\top U = I_{q \times q}$, and $V$ is $n \times r$, with $r \leq n$ and $V^\top V = I_{r \times r}$. Suppose the matrix $B$ has a column space spanned by $U$. If $span(U) \subseteq span(V))$*

$$VV^\top B = B \tag{37}$$

*Proof.* Because $B$ has a column space spanned by $U$, we can write $B = UR$ for some matrix $R$. Furthermore, because $\text{span}(U) \subseteq \text{span}(V))$, we may write $U = VT$, for some $r \times q$ matrix $T$, with $q \leq r$. Indeed we know that $T$ has orthonormal columns, since $U^\top U = T^\top V^\top VT = T^\top T = I_{q \times q}$. Hence, we can write $B = VTR$, where $T$ is some orthonormal matrix. Now

$$VV^\top B = VV^\top VTR = VTR = B. \tag{38}$$

$\square$

### 3.1 Proof of Theorem 1

**Theorem 1.** *Suppose that the conditions of Lemma 1 are met for a fixed instance of $X, \beta^*$. If $\mathrm{span}(U_S) \subseteq \mathrm{span}(U_\mathcal{A})$, then after preconditioning using HJ the conditions continue to hold, and*

$$\frac{\lambda_u}{\lambda_l} \preceq \frac{\bar{\lambda}_u}{\bar{\lambda}_l}, \tag{39}$$

*where the stochasticity on both sides is due to independent noise vectors $w$. On the other hand, if $X_S^\top P_X^\top P_X X_S$ is not invertible then HJ cannot in general recover the signed support.*

*Proof.* We have $P_X = P_y = U_\mathcal{A} U_\mathcal{A}^\top$. With this, $\bar{w} = U_\mathcal{A} U_\mathcal{A}^\top w$. First, consider the case that $\mathrm{span}(U_S) \subseteq \mathrm{span}(U_\mathcal{A})$. Using Lemmas 1 and 2 we have

$$\bar{\mu}_j = X_j^\top U_\mathcal{A} U_\mathcal{A}^\top U_\mathcal{A} U_\mathcal{A}^\top X_S (X_S^\top U_\mathcal{A} U_\mathcal{A}^\top U_\mathcal{A} U_\mathcal{A}^\top X_S)^{-1} \mathrm{sgn}(\beta_S^*) \tag{40}$$

$$= X_j^\top X_S (X_S^\top X_S)^{-1} \mathrm{sgn}(\beta_S^*) = \mu_j \tag{41}$$

$$\bar{\gamma}_i = e_i^\top \left( \frac{1}{n} X_S^\top U_\mathcal{A} U_\mathcal{A}^\top U_\mathcal{A} U_\mathcal{A}^\top X_S \right)^{-1} \mathrm{sgn}(\beta_S^*) \tag{42}$$

$$= e_i^\top \left( \frac{1}{n} X_S^\top X_S \right)^{-1} \mathrm{sgn}(\beta_S^*) = \gamma_i \tag{43}$$

$$\bar{\eta}_j = X_j^\top U_\mathcal{A} U_\mathcal{A}^\top \left( I_{n\times n} - U_\mathcal{A} U_\mathcal{A}^\top X_S (X_S^\top U_\mathcal{A} U_\mathcal{A}^\top U_\mathcal{A} U_\mathcal{A}^\top X_S)^{-1} X_S^\top U_\mathcal{A} U_\mathcal{A}^\top \right) U_\mathcal{A} U_\mathcal{A}^\top \frac{w}{n} \tag{44}$$

$$= X_j^\top U_\mathcal{A} U_\mathcal{A}^\top \left( I_{n\times n} - X_S \left( X_S^\top X_S \right)^{-1} X_S^\top \right) U_\mathcal{A} U_\mathcal{A}^\top \frac{w}{n} \tag{45}$$

$$= X_j^\top U_\mathcal{A} U_\mathcal{A}^\top \left( I_{n\times n} - U_S U_S^\top \right) U_\mathcal{A} U_\mathcal{A}^\top \frac{w}{n} \tag{46}$$

$$= X_j^\top \left( U_\mathcal{A} U_\mathcal{A}^\top - U_S U_S^\top U_\mathcal{A} U_\mathcal{A}^\top \right) \frac{w}{n} \tag{47}$$

$$= X_j^\top \left( I_{n\times n} - U_S U_S^\top \right) U_\mathcal{A} U_\mathcal{A}^\top \frac{w}{n} \tag{48}$$

$$\bar{\epsilon}_i = e_i^\top \left( \frac{1}{n} X_S^\top U_\mathcal{A} U_\mathcal{A}^\top U_\mathcal{A} U_\mathcal{A}^\top X_S \right)^{-1} X_S^\top U_\mathcal{A} U_\mathcal{A}^\top U_\mathcal{A} U_\mathcal{A}^\top \frac{w}{n} \tag{49}$$

$$= e_i^\top \left( \frac{1}{n} X_S^\top X_S \right)^{-1} X_S^\top \frac{w}{n} = \epsilon_i. \tag{50}$$

We immediately see that if the conditions of Lemma 1 hold for the original problem (i.e., $X_S^\top X_S$ is invertible, $|\mu_j| < 1 \;\; \forall j \in S^c$ and $\mathrm{sgn}(\beta_i^*)\gamma_i > 0 \;\; \forall i \in S$), they continue to hold after preconditioning using HJ (i.e., $\bar{X}_S^\top \bar{X}_S$ is invertible, $|\bar{\mu}_j| < 1 \;\; \forall j \in S^c$ and $\mathrm{sgn}(\beta_i^*)\bar{\gamma}_i > 0 \;\; \forall i \in S$). Furthermore, we have $\bar{\lambda}_u = \lambda_u$. Next, we must show that $\bar{\lambda}_l \preceq \lambda_l$. We will simplify this task as follows. Note that

$$\bar{\lambda}_l = \max_{j \in S^c} \frac{\bar{\eta}_j}{(2[\![\bar{\eta}_j > 0]\!] - 1) - \bar{\mu}_j} = \max \left( \max_{j \in S^c} \frac{\bar{\eta}_j}{-1 - \bar{\mu}_j}, \max_{j \in S^c} \frac{\bar{\eta}_j}{1 - \bar{\mu}_j} \right) \tag{51}$$

$$= \max \left( \left\{ \frac{\bar{\eta}_j}{-1 - \bar{\mu}_j}, \frac{\bar{\eta}_j}{1 - \bar{\mu}_j} \right\}_{j \in S^c} \right) \tag{52}$$

$$\lambda_l = \max_{j \in S^c} \frac{\eta_j}{(2[\![\eta_j > 0]\!] - 1) - \mu_j} = \max \left( \max_{j \in S^c} \frac{\eta_j}{-1 - \mu_j}, \max_{j \in S^c} \frac{\eta_j}{1 - \mu_j} \right) \tag{53}$$

$$= \max \left( \left\{ \frac{\eta_j}{-1 - \mu_j}, \frac{\eta_j}{1 - \mu_j} \right\}_{j \in S^c} \right) \tag{54}$$

where the $\bar{\mu}_j = \mu_j$ are fixed because $X, \beta^*$ are fixed. By our derivation in Eq. (48), the effect of preconditioning on $\eta_j$ can be viewed as further restricting the subspace in which the noise $w$ lies, while keeping $X_j$ and $\mu_j$ fixed. Specifically, in $\eta_j$, $w$ is pre-multiplied by $\left( I_{n\times n} - U_S U_S^\top \right)$, while in $\bar{\eta}_j$ it is pre-multiplied by $\left( I_{n\times n} - U_S U_S^\top \right) U_\mathcal{A} U_\mathcal{A}^\top$. Whatever $U_\mathcal{A}$, the latter projection eliminates

at least as large a subspace as the former. Because the $X_j$ and $\bar{\mu}_j = \mu_j$ are fixed, it follows by symmetry of the Gaussian that

$$\bar{\lambda}_l = \max\left(\left\{\frac{\bar{\eta}_j}{-1-\bar{\mu}_j}, \frac{\bar{\eta}_j}{1-\bar{\mu}_j}\right\}_{j \in S^c}\right) \preceq \max\left(\left\{\frac{\eta_j}{-1-\mu_j}, \frac{\eta_j}{1-\mu_j}\right\}_{j \in S^c}\right) = \lambda_l, \quad (55)$$

where the stochasticity is due to the noise $w$. Rewriting some of the variables, we observe that $\bar{\lambda}_l$ and $\lambda_l$ are both independent of $\bar{\lambda}_u = \lambda_u$. Specifically, if $\text{span}(U_S) \subseteq \text{span}(U_{\mathcal{A}})$ then using Lemma 2

$$\eta_j = \frac{1}{n}X_j^\top \left(I_{n \times n} - U_S U_S^\top\right) w \quad (56)$$

$$\bar{\eta}_j = \frac{1}{n}X_j^\top \left(I_{n \times n} - U_S U_S^\top\right) U_{\mathcal{A}} U_{\mathcal{A}}^\top w \quad (57)$$

$$\epsilon_i = \frac{1}{n}e_i^\top \left(\frac{1}{n}X_S^\top X_S\right)^{-1} X_S^\top U_S U_S^\top w \quad (58)$$

$$= \bar{\epsilon}_i = \frac{1}{n}e_i^\top \left(\frac{1}{n}X_S^\top X_S\right)^{-1} X_S^\top U_S U_S^\top U_{\mathcal{A}} U_{\mathcal{A}}^\top w \quad (59)$$

Since the variables $\left(I_{n \times n} - U_S U_S^\top\right) w$ and $U_S U_S^\top w$ are jointly Gaussian distributed with zero covariance, they are independent. Thus, $\eta_j$ and $\epsilon_i = \bar{\epsilon}_i$ are independent, and because randomness is only due to the noise $w$, therefore also $\lambda_l$ and $\lambda_u = \bar{\lambda}_u$. By the same reasoning, $\left(I_{n \times n} - U_S U_S^\top\right) U_{\mathcal{A}} U_{\mathcal{A}}^\top w$ and $U_S U_S^\top U_{\mathcal{A}} U_{\mathcal{A}}^\top w$ are independent. This in turn implies that $\bar{\lambda}_l$ and $\bar{\lambda}_u = \lambda_u$ are independent. We now combine these results: Recall that we defined $1/\bar{\lambda}_l = \infty$ and $1/\lambda_l = \infty$ if $\bar{\lambda}_l = 0$ or $\lambda_l = 0$. Because $\bar{\lambda}_l \preceq \lambda_l$ and $\bar{\lambda}_l \geq 0$, $\lambda_l \geq 0$, we have that $1/\lambda_l \preceq 1/\bar{\lambda}_l$. Next, because both $1/\bar{\lambda}_l, 1/\lambda_l$ are independent of $\bar{\lambda}_u = \lambda_u \geq 0$, we have

$$\frac{\lambda_u}{\lambda_l} \preceq \frac{\bar{\lambda}_u}{\bar{\lambda}_l}. \quad (60)$$

On the other hand, if $X_S^\top P_X^\top P_X X_S$ is not invertible, the conditions of Lemma 1 are not met, and so signed support recovery is in general not possible. $\quad\square$

## 3.2 Proof of Theorem 2

**Theorem 2.** *Suppose that the conditions of Lemma 1 are met for a fixed instance of $X, \beta^*$. If $span(U_S) \subseteq span(U_A)$, then after preconditioning using PBHT the conditions continue to hold, and*

$$\frac{\lambda_u}{\lambda_l} \preceq \frac{\bar{\lambda}_u}{\bar{\lambda}_l}, \tag{61}$$

*where the stochasticity on both sides is due to independent noise vectors $w$. On the other hand, if $span(U_{S^c}) = span(U_A)$, then PBHT cannot recover the signed support.*

*Proof.* We have $P_X = I_{n \times n}, P_y = U_A U_A^\top$. With this, $\bar{w} = (U_A U_A^\top - I_{n \times n})X\beta^* + U_A U_A^\top w$. Now let us consider the case that $span(U_S) \subseteq span(U_A)$. Using Lemma 2 we have

$$\bar{\mu}_j = X_j^\top X_S (X_S^\top X_S)^{-1} \text{sgn}(\beta_S^*) = \mu_j \tag{62}$$

$$\bar{\gamma}_i = e_i^\top \left(\frac{1}{n}X_S^\top X_S\right)^{-1} \text{sgn}(\beta_S^*) = \gamma_i \tag{63}$$

$$\bar{\eta}_j = \frac{1}{n}X_j^\top \left(I_{n \times n} - U_S U_S^\top\right)\left((U_A U_A^\top - I_{n \times n})X\beta^* + U_A U_A^\top w\right) \tag{64}$$

$$= X_j^\top \left(I_{n \times n} - U_S U_S^\top\right)U_A U_A^\top \frac{w}{n} \tag{65}$$

$$\bar{\epsilon}_i = \frac{1}{n}e_i^\top \left(\frac{1}{n}X_S^\top X_S\right)^{-1}X_S^\top \left((U_A U_A^\top - I_{n \times n})X\beta^* + U_A U_A^\top w\right) \tag{66}$$

$$= \frac{1}{n}e_i^\top \left(\frac{1}{n}X_S^\top X_S\right)^{-1}X_S^\top U_A U_A^\top w = \epsilon_i. \tag{67}$$

Since $P_X = I_{n \times n}$, we immediately see that if the conditions of Lemma 1 hold for the original problem, they continue to hold after preconditioning using PBHT. Furthermore, we see that $\bar{\lambda}_u = \lambda_u$. Next, we must show that $\bar{\lambda}_l \preceq \lambda_l$. We will approach this task in a similar manner as in Theorem 1. For completeness we repeat the main steps here. Note that

$$\bar{\lambda}_l = \max\left(\left\{\frac{\bar{\eta}_j}{-1-\bar{\mu}_j}, \frac{\bar{\eta}_j}{1-\bar{\mu}_j}\right\}_{j \in S^c}\right) \qquad \lambda_l = \max\left(\left\{\frac{\eta_j}{-1-\mu_j}, \frac{\eta_j}{1-\mu_j}\right\}_{j \in S^c}\right). \tag{68}$$

As before, the effect of preconditioning on $\eta_j$ can be viewed as further restricting the subspace in which the noise $w$ lies, while keeping $X_j$ and $\mu_j$ fixed. Specifically, in $\eta_j$, $w$ is pre-multiplied by $\left(I_{n \times n} - U_S U_S^\top\right)$, while in $\bar{\eta}_j$ it is pre-multiplied by $\left(I_{n \times n} - U_S U_S^\top\right)U_A U_A^\top$. Whatever $U_A$, the latter projection eliminates at least as large a subspace as the former and so because the $X_j$ and $\bar{\mu}_j = \mu_j$ are fixed, it follows that

$$\bar{\lambda}_l = \max\left(\left\{\frac{\bar{\eta}_j}{-1-\bar{\mu}_j}, \frac{\bar{\eta}_j}{1-\bar{\mu}_j}\right\}_{j \in S^c}\right) \preceq \max\left(\left\{\frac{\eta_j}{-1-\mu_j}, \frac{\eta_j}{1-\mu_j}\right\}_{j \in S^c}\right) = \lambda_l, \tag{69}$$

where the stochasticity is due to the noise $w$. The remaining part of the theorem again mirrors that of Theorem 1, which we repeat here for completeness. Rewriting some of the variables we observe that $\bar{\lambda}_l$ and $\lambda_l$ are both independent of $\bar{\lambda}_u = \lambda_u$. Specifically, if $span(U_S) \subseteq span(U_A)$ then using Lemma 2

$$\eta_j = \frac{1}{n}X_j^\top \left(I_{n \times n} - U_S U_S^\top\right)w \tag{70}$$

$$\bar{\eta}_j = \frac{1}{n}X_j^\top \left(I_{n \times n} - U_S U_S^\top\right)U_A U_A^\top w \tag{71}$$

$$\epsilon_i = \frac{1}{n}e_i^\top \left(\frac{1}{n}X_S^\top X_S\right)^{-1}X_S^\top U_S U_S^\top w \tag{72}$$

$$= \bar{\epsilon}_i = \frac{1}{n}e_i^\top \left(\frac{1}{n}X_S^\top X_S\right)^{-1}X_S^\top U_S U_S^\top U_A U_A^\top w \tag{73}$$

Since $\left(I_{n\times n} - U_S U_S^\top\right) w$ and $U_S U_S^\top w$ are jointly Gaussian with zero covariance, they are independent. Thus, $\eta_j$ and $\epsilon_i = \bar{\epsilon}_i$ are independent and so are $\lambda_l$ and $\lambda_u = \bar{\lambda}_u$. By similar reasoning, $\left(I_{n\times n} - U_S U_S^\top\right) U_\mathcal{A} U_\mathcal{A}^\top w$ and $U_S U_S^\top U_\mathcal{A} U_\mathcal{A}^\top w$ are independent, hence so are $\bar{\lambda}_l$ and $\bar{\lambda}_u = \lambda_u$. We now combine these results: Because $\bar{\lambda}_l \preceq \lambda_l$ and $\bar{\lambda}_l \geq 0$, $\lambda_l \geq 0$, we have that $1/\lambda_l \preceq 1/\bar{\lambda}_l$. Next, because both $1/\bar{\lambda}_l, 1/\lambda_l$ are independent of $\bar{\lambda}_u = \lambda_u \geq 0$, we have

$$\frac{\lambda_u}{\lambda_l} \preceq \frac{\bar{\lambda}_u}{\bar{\lambda}_l}, \tag{74}$$

On the other hand, if $\text{span}(U_{S^c}) = \text{span}(U_\mathcal{A})$

$$\bar{\mu}_j = X_j^\top X_S (X_S^\top X_S)^{-1} \text{sgn}(\beta_S^*) = \mu_j \tag{75}$$

$$\bar{\gamma}_i = e_i^\top \left(\frac{1}{n} X_S^\top X_S\right)^{-1} \text{sgn}(\beta_S^*) = \gamma_i \tag{76}$$

$$\bar{\eta}_j = \frac{1}{n} X_j^\top \left(I_{n\times n} - U_S U_S^\top\right) \left((U_\mathcal{A} U_\mathcal{A}^\top - I_{n\times n}) X\beta^* + U_\mathcal{A} U_\mathcal{A}^\top w\right) \tag{77}$$

$$= \frac{1}{n} X_j^\top \left(I_{n\times n} - U_S U_S^\top\right) w = \eta_j \tag{78}$$

$$\bar{\epsilon}_i = \frac{1}{n} e_i^\top \left(\frac{1}{n} X_S^\top X_S\right)^{-1} X_S^\top \left((U_\mathcal{A} U_\mathcal{A}^\top - I_{n\times n}) X\beta^* + U_\mathcal{A} U_\mathcal{A}^\top w\right) \tag{79}$$

$$= \frac{1}{n} e_i^\top \left(\frac{1}{n} X_S^\top X_S\right)^{-1} X_S^\top \left(U_\mathcal{A} U_\mathcal{A}^\top w - X\beta^*\right) \tag{80}$$

$$= -e_i^\top \left(X_S^\top X_S\right)^{-1} X_S^\top X\beta^* \tag{81}$$

$$= -e_i^\top \left(X_S^\top X_S\right)^{-1} X_S^\top X_S \beta_S^* \tag{82}$$

$$= -\beta_i^* \tag{83}$$

Thus the conditions of Lemma 1 continue to hold and we have $\bar{\lambda}_l = \lambda_l$ and

$$\bar{\lambda}_u = \min_{i \in S} \left|\frac{\beta_i^* + \bar{\epsilon}_i}{\bar{\gamma}_i}\right|_+ = \min_{i \in S} \left|\frac{\beta_i^* - \beta_i^*}{\bar{\gamma}_i}\right|_+ = 0 \tag{84}$$

Recall that in Section 3.2 of the main paper we defined $\bar{\lambda}_u/\bar{\lambda}_l \triangleq 0$ if $\bar{\lambda}_u = \bar{\lambda}_l = 0$. Because $\bar{\lambda}_l$ is with probability 1 non-negative, this means that with probability 1, $\bar{\lambda}_u/\bar{\lambda}_l = 0$ and signed support recovery is not possible. $\qquad\square$

# 4 Proofs of Section 5

**Lemma 3.** *Assume that the spectra $\Sigma_S, \Sigma_{S^c}$ are derived by normalizing unconstrained spectra $\hat{\Sigma}_S$ and $\hat{\Sigma}_{S^c}$ as*

$$\Sigma_S = \frac{\hat{\Sigma}_S}{\|\hat{\Sigma}_S\|_F} \sqrt{kn} \tag{85}$$

$$\Sigma_{S^c} = \frac{\hat{\Sigma}_{S^c}}{\|\hat{\Sigma}_{S^c}\|_F} \sqrt{(p-k)n}. \tag{86}$$

*Then the squared column norms of $X$ are on expectation $n$.*

*Proof.* We have $\forall i \in S$,

$$E(X_i^\top X_i) = E(v_{S,i,\cdot} \Sigma_S^\top U^\top U \Sigma_S v_{S,i,\cdot}^\top) \tag{87}$$

$$= kn E\left( v_{S,i,\cdot} \frac{\hat{\Sigma}_S^\top \hat{\Sigma}_S}{\|\hat{\Sigma}_S\|_F^2} v_{S,i,\cdot}^\top \right) \tag{88}$$

$$= kn \sum_{i'=1}^{k} E\left( v_{S,i,i'}^2 \right) \frac{\hat{\sigma}_{S,i'}^2}{\|\hat{\Sigma}_S\|_F^2} = n, \tag{89}$$

and $\forall j \in S^c$,

$$E(X_j^\top X_j) = E(v_{S^c,j-k,\cdot} \Sigma_{S^c}^\top U^\top U \Sigma_{S^c} v_{S^c,j-k,\cdot}^\top) \tag{90}$$

$$= (p-k)n E\left( v_{S^c,j-k,\cdot} \frac{\hat{\Sigma}_{S^c}^\top \hat{\Sigma}_{S^c}}{\|\hat{\Sigma}_{S^c}\|_F^2} v_{S^c,j-k,\cdot}^\top \right) \tag{91}$$

$$= (p-k)n \sum_{j'=1}^{p-k} E\left( v_{S^c,j-k,j'}^2 \right) \frac{\hat{\sigma}_{S^c,j'}^2}{\|\hat{\Sigma}_{S^c}\|_F^2} = n. \tag{92}$$

$\square$

## 4.1 Proof of Theorem 3

**Theorem 3.** *Assume the Lasso problem was generated according to the generative model of Section 5.1 in the main paper with $\forall i \in \sigma(S), \hat{\sigma}_{S,i} = 1, \hat{\sigma}_{S^c,i} = \kappa$ and $\forall j \in \sigma(S^c), \hat{\sigma}_{S^c,j} = 1$ and that $\kappa < \sqrt{n-k}/\sqrt{k(p-k-1)}$. Then the conditions of Lemma 1 hold before and after preconditioning using JR. Moreover,*

$$\frac{\bar{\lambda}_u}{\bar{\lambda}_l} = \frac{(p-k)}{n+p\kappa^2-k}\frac{\lambda_u}{\lambda_l}. \tag{93}$$

*Proof.* Normalizing $\hat{\Sigma}_S$ and $\hat{\Sigma}_{S^c}$ to yield $\Sigma_S, \Sigma_{S^c}$, as required by the model for $X$,

$$\sigma_{S,i} = \frac{\sqrt{kn}}{\sqrt{k}} = \sqrt{n} \qquad \forall i \in \sigma(S) \tag{94}$$

$$\sigma_{S^c,i} = \frac{\sqrt{n(p-k)}\kappa}{\sqrt{k\kappa^2+n-k}} \quad \forall i \in \sigma(S) \qquad \sigma_{S^c,j} = \frac{\sqrt{n(p-k)}}{\sqrt{k\kappa^2+n-k}} \quad \forall j \in \sigma(S^c). \tag{95}$$

Because $\Sigma_S$ has constant spectrum, it is easy to see that $X_S^\top X_S = cI_{k\times k}$, for some $c > 0$. This means that $X_S^\top X_S$ is invertible and $\text{sgn}(\beta_i^*)\gamma_i > 0$. Let's look at the variables $\mu_j$:

$$|\mu_j| = \left|X_j^\top X_S (X_S^\top X_S)^{-1}\text{sgn}(\beta_S^*)\right| \tag{96}$$

$$= \left|v_{S^c,j-k,\cdot}\Sigma_{S^c}^\top U^\top U\Sigma_S V_S^\top (V_S\Sigma_S^\top U^\top U\Sigma_S V_S^\top)^{-1}\text{sgn}(\beta_S^*)\right| \tag{97}$$

$$= \left|v_{S^c,j-k,\cdot}\Sigma_{S^c}^\top \Sigma_S V_S^\top V_S(\Sigma_S^\top \Sigma_S)^{-1}V_S^\top \text{sgn}(\beta_S^*)\right| \tag{98}$$

$$= \left|v_{S^c,j-k,\cdot}\Sigma_{S^c}^\top \Sigma_S(\Sigma_S^\top \Sigma_S)^{-1}V_S^\top \text{sgn}(\beta_S^*)\right| \tag{99}$$

$$= \left|\left[v_{S^c,j-k,\cdot}\Sigma_{S^c}^\top\right]_{(1:k)} \Sigma_{S,(1:k),(1:k)}^{-1}V_S^\top \text{sgn}(\beta_S^*)\right| \tag{100}$$

$$= \left|\left[v_{S^c,j-k,\cdot}\frac{\sqrt{n(p-k)}\kappa}{\sqrt{k\kappa^2+n-k}}\right]_{(1:k)} \frac{1}{\sqrt{n}}V_S^\top \text{sgn}(\beta_S^*)\right| \tag{101}$$

$$= \frac{\sqrt{(p-k)}\kappa}{\sqrt{k\kappa^2+n-k}} \left|[v_{S^c,j-k,\cdot}]_{(1:k)} V_S^\top \text{sgn}(\beta_S^*)\right| \tag{102}$$

$$\overset{\text{Cauchy}}{\leq} \frac{\sqrt{(p-k)}\kappa}{\sqrt{k\kappa^2+n-k}} \left\|V_S[v_{S^c,j-k,\cdot}]_{(1:k)}^\top\right\|_2 \|\text{sgn}(\beta_S^*)\|_2 \tag{103}$$

$$= \frac{\sqrt{k(p-k)}\kappa}{\sqrt{k\kappa^2+n-k}} \left\|V_S[v_{S^c,j-k,\cdot}]_{(1:k)}^\top\right\|_2 \tag{104}$$

$$\leq \frac{\sqrt{k(p-k)}\kappa}{\sqrt{k\kappa^2+n-k}} \|V_S\|_2 \left\|[v_{S^c,j-k,\cdot}]_{(1:k)}\right\|_2 \tag{105}$$

$$\leq \frac{\sqrt{k(p-k)}\kappa}{\sqrt{k\kappa^2+n-k}}. \tag{106}$$

Because $\kappa < \sqrt{(n-k)/(k(p-k-1))}$,

$$\frac{\sqrt{k(p-k)}\kappa}{\sqrt{k\kappa^2+n-k}} < \frac{\sqrt{k(p-k)}\sqrt{\frac{n-k}{k(p-k-1)}}}{\sqrt{k\frac{n-k}{k(p-k-1)}+n-k}} \tag{107}$$

$$= \frac{\sqrt{\frac{(p-k)(n-k)}{p-k-1}}}{\sqrt{\frac{n-k+(n-k)(p-k-1)}{p-k-1}}} = \frac{\sqrt{\frac{(p-k)(n-k)}{p-k-1}}}{\sqrt{\frac{(n-k)(p-k)}{p-k-1}}} = 1, \tag{108}$$

and so the conditions of Lemma 1 are met. We can then apply Lemma 1 and simplify the resulting upper and lower bounds $\lambda_u, \lambda_l$ on $\lambda$. Plugging in $\Sigma_S$ and $\Sigma_{S^c}$ we see that the data matrix $X$ satisfies

$$
\begin{align}
XX^\top &= U\left[\Sigma_S V_S^\top, \Sigma_{S^c} V_{S^c}^\top\right]\left[\Sigma_S V_S^\top, \Sigma_{S^c} V_{S^c}^\top\right]^\top U^\top \tag{109}\\
&= U\left[\Sigma_S \Sigma_S^\top + \Sigma_{S^c}\Sigma_{S^c}^\top\right]U^\top \tag{110}\\
&\triangleq UDD^\top U^\top. \tag{111}
\end{align}
$$

From this we see that $X = UDV^\top$ has left eigenvectors $U$ and singular values

$$
\begin{align}
d_i &= \sqrt{\sigma_{S,i}^2 + \sigma_{S^c,i}^2} = \sqrt{n + \frac{n(p-k)\kappa^2}{k\kappa^2 + n - k}} \quad \forall i \in \sigma(S) \tag{112}\\
d_j &= \sqrt{\frac{n(p-k)}{k\kappa^2 + n - k}} \quad \forall j \in \sigma(S^c). \tag{113}
\end{align}
$$

Recall that for JR, $P_X = P_y = U\left(DD^\top\right)^{-1/2} U^\top$. After projecting, we find that

$$
\begin{align}
\bar{\mu}_j &= X_j^\top P_X^\top P_X X_S (X_S^\top P_X^\top P_X X_S)^{-1}\mathrm{sgn}(\beta_S^*) \tag{114}\\
&= \left|v_{S^c,j-k,\cdot}\Sigma_{S^c}^\top U^\top P_X^\top P_X U\Sigma_S V_S^\top (V_S\Sigma_S^\top U^\top P_X^\top P_X U\Sigma_S V_S^\top)^{-1}\mathrm{sgn}(\beta_S^*)\right| \tag{115}\\
&= \left|v_{S^c,j-k,\cdot}\Sigma_{S^c}^\top \left(DD^\top\right)^{-1}\Sigma_S V_S^\top \left(V_S\Sigma_S^\top \left(DD^\top\right)^{-1}\Sigma_S V_S^\top\right)^{-1}\mathrm{sgn}(\beta_S^*)\right| \tag{116}\\
&= \left|v_{S^c,j-k,\cdot}\Sigma_{S^c}^\top \left(DD^\top\right)^{-1}\Sigma_S \left(\Sigma_S^\top \left(DD^\top\right)^{-1}\Sigma_S\right)^{-1}V_S^\top \mathrm{sgn}(\beta_S^*)\right| \tag{117}\\
&= \left|v_{S^c,j-k,\cdot}\Sigma_{S^c}^\top \Sigma_S \left(\Sigma_S^\top \Sigma_S\right)^{-1}V_S^\top \mathrm{sgn}(\beta_S^*)\right| = \mu_j \tag{118}\\
\bar{\gamma}_i &= e_i^\top \left(\frac{1}{n}X_S^\top P_X^\top P_X X_S\right)^{-1}\mathrm{sgn}(\beta_S^*) \tag{119}\\
&= \left(n + \frac{n(p-k)\kappa^2}{k\kappa^2 + n - k}\right)e_i^\top \left(\frac{1}{n}X_S^\top X_S\right)^{-1}\mathrm{sgn}(\beta_S^*) \tag{120}\\
&= \left(n + \frac{n(p-k)\kappa^2}{k\kappa^2 + n - k}\right)\gamma_i \tag{121}
\end{align}
$$

$$\bar{\eta}_j = X_j^\top P_X^\top \left( I_{n \times n} - P_X X_S (X_S^\top P_X^\top P_X X_S)^{-1} X_S^\top P_X^\top \right) \frac{\bar{w}}{n} \tag{122}$$

$$= v_{S^c, j-k, .} \Sigma_{S^c}^\top U^\top P_X^\top$$
$$\left( I_{n \times n} - P_X U \Sigma_S V_S^\top (V_S \Sigma_S^\top U^\top P_X^\top P_X U \Sigma_S V_S^\top)^{-1} V_S \Sigma_S^\top U^\top P_X^\top \right) \frac{P_X w}{n} \tag{123}$$

$$= v_{S^c, j-k, .} \Sigma_{S^c}^\top \left( DD^\top \right)^{-1/2} U^\top$$
$$\left( I_{n \times n} - U \left( DD^\top \right)^{-1/2} \Sigma_S \left( \Sigma_S^\top \left( DD^\top \right)^{-1} \Sigma_S \right)^{-1} \Sigma_S^\top \left( DD^\top \right)^{-1/2} U^\top \right) \frac{P_X w}{n} \tag{124}$$

$$= v_{S^c, j-k, .} \Sigma_{S^c}^\top \left( DD^\top \right)^{-1/2} U^\top \left( I_{n \times n} - U_S U_S^\top \right) \frac{P_X w}{n} \tag{125}$$

$$= v_{S^c, j-k, .} \Sigma_{S^c}^\top \left( DD^\top \right)^{-1/2} \left( U^\top - \begin{bmatrix} I_{k \times k} \\ 0 \end{bmatrix} U_S^\top \right) \frac{P_X w}{n} \tag{126}$$

$$= v_{S^c, j-k, .} \Sigma_{S^c}^\top \left( DD^\top \right)^{-1/2} \begin{bmatrix} 0 \\ U_{S^c}^\top \end{bmatrix} \frac{P_X w}{n} \tag{127}$$

$$= \left[ v_{S^c, j-k, .} \Sigma_{S^c}^\top \left( DD^\top \right)^{-1/2} \right]_{(k+1:n)} U_{S^c}^\top \frac{P_X w}{n} \tag{128}$$

$$= \left[ v_{S^c, j-k, .} \Sigma_{S^c}^\top \left( DD^\top \right)^{-1/2} \right]_{(k+1:n)} \begin{bmatrix} 0 & I_{n-k \times n-k} \end{bmatrix} \left( DD^\top \right)^{-1/2} U^\top \frac{w}{n} \tag{129}$$

$$= \left[ v_{S^c, j-k, .} \Sigma_{S^c}^\top \left( DD^\top \right)^{-1/2} \right]_{(k+1:n)} \begin{bmatrix} 0 & D_{(k+1:n),(k+1:n)}^{-1} \end{bmatrix} U^\top \frac{w}{n} \tag{130}$$

$$= \left[ v_{S^c, j-k, .} \Sigma_{S^c}^\top \left( DD^\top \right)^{-1/2} \right]_{(k+1:n)} D_{(k+1:n),(k+1:n)}^{-1} U_{S^c}^\top \frac{w}{n} \tag{131}$$

$$= \left[ v_{S^c, j-k, .} \Sigma_{S^c}^\top \right]_{(k+1:n)} D_{(k+1:n),(k+1:n)}^{-2} U_{S^c}^\top \frac{w}{n} \tag{132}$$

$$= \frac{1}{n(p-k)/(k\kappa^2 + n - k)} \left[ v_{S^c, j-k, .} \Sigma_{S^c}^\top \right]_{(k+1:n)} U_{S^c}^\top \frac{w}{n} \tag{133}$$

$$= \frac{1}{n(p-k)/(k\kappa^2 + n - k)} \eta_j \tag{134}$$

$$\bar{\epsilon}_i = e_i^\top \left( \frac{1}{n} X_S^\top P_X^\top P_X X_S \right)^{-1} X_S^\top P_X^\top P_X \frac{w}{n} \tag{135}$$

$$= e_i^\top \left( \frac{1}{n} X_S^\top X_S \right)^{-1} X_S^\top \frac{w}{n} = \epsilon_i \tag{136}$$

$$\tag{137}$$

Immediately we see that the conditions of Lemma 1 continue to hold after preconditioning using JR. Note that by the above derivation $(2[\![\bar{\eta}_j > 0]\!] - 1) - \bar{\mu}_j = (2[\![\eta_j > 0]\!] - 1) - \mu_j$, and so

$$\bar{\lambda}_l = \max_{j \in S^c} \frac{\bar{\eta}_j}{(2[\![\bar{\eta}_j > 0]\!] - 1) - \bar{\mu}_j} = \frac{1}{n(p-k)/(k\kappa^2 + n - k)} \max_{j \in S^c} \frac{\eta_j}{(2[\![\eta_j > 0]\!] - 1) - \mu_j} \tag{138}$$

$$= \frac{1}{n(p-k)/(k\kappa^2 + n - k)} \lambda_l \tag{139}$$

$$\bar{\lambda}_u = \min_{i \in S} \left| \frac{\beta_i^* + \bar{\epsilon}_i}{\bar{\gamma}_i} \right|_+ = \frac{1}{n + (n(p-k)\kappa^2/(k\kappa^2 + n - k))} \min_{i \in S} \left| \frac{\beta_i^* + \epsilon_i}{\gamma_i} \right|_+ \tag{140}$$

$$= \frac{1}{n + (n(p-k)\kappa^2/(k\kappa^2 + n - k))} \lambda_u. \tag{141}$$

The new ratio $\bar{\lambda}_u/\bar{\lambda}_l$ of upper and lower bounds then becomes

$$\frac{\bar{\lambda}_u}{\bar{\lambda}_l} = \frac{n(p-k)/(k\kappa^2 + n - k)}{n + (n(p-k)\kappa^2/(k\kappa^2 + n - k))} \frac{\lambda_u}{\lambda_l} \tag{142}$$

$$= \frac{n(p-k)}{n(k\kappa^2 + n - k) + n(p-k)\kappa^2} \frac{\lambda_u}{\lambda_l} \tag{143}$$

$$= \frac{p-k}{(k\kappa^2 + n - k) + (p-k)\kappa^2} \frac{\lambda}{\lambda_l} \tag{144}$$

$$= \frac{p-k}{n + p\kappa^2 - k} \frac{\lambda_u}{\lambda_l}. \tag{145}$$

$$\square$$

## 4.2 Gaussian Designs with Piecewise Constant Spectra

The generative model presented in Section 5.1 of the paper uses an *orthonormal* column basis $U$ to generate $X$. The question arises whether a more natural Gaussian design $X$ exists that is in a sense equivalent to the orthonormal construction of Section 5.1. In this section we present a generative model that uses a Gaussian column "basis" that achieves this. As before, let $V_S$ and $V_{S^c}$ be random orthonormal bases of sizes $k \times k$ and $p - k \times p - k$ respectively and let $\Sigma_S$ and $\Sigma_{S^c}$ be rectangular matrices that are derived from matrices $\hat{\Sigma}_S, \hat{\Sigma}_{S^c}$ as in Section 5.1 of the paper. Let $W^m$ be an $m \times n$ matrix of independent Gaussians with marginal distribution $\mathcal{N}(0, 1)$. Then we let

$$X^m = \frac{1}{\sqrt{n}} W^m \left[ \Sigma_S V_S^\top, \Sigma_{S^c} V_{S^c}^\top \right]. \tag{146}$$

We note that all columns of $X$ are mean zero, and their squared norms are on expectation $m$:

$$E(X_i^m) = E(Xe_i) = \frac{1}{\sqrt{n}} E(W^m \left[ \Sigma_S V_S^\top, \Sigma_{S^c} V_{S^c}^\top \right] e_i) \tag{147}$$

$$= \frac{1}{\sqrt{n}} E(W^m) E(\left[ \Sigma_S V_S^\top, \Sigma_{S^c} V_{S^c}^\top \right] e_i) = 0 \tag{148}$$

$$E(X_i^{m\top} X_i^m) = E(e_i^\top X^\top X e_i) \tag{149}$$

$$= \frac{1}{n} E(e_i^\top \left[ \Sigma_S V_S^\top, \Sigma_{S^c} V_{S^c}^\top \right]^\top W^{m\top} W^m \left[ \Sigma_S V_S^\top, \Sigma_{S^c} V_{S^c}^\top \right] e_i) \tag{150}$$

$$= \frac{m}{n} E(e_i^\top \left[ \Sigma_S V_S^\top, \Sigma_{S^c} V_{S^c}^\top \right]^\top \left[ \Sigma_S V_S^\top, \Sigma_{S^c} V_{S^c}^\top \right] e_i) \tag{151}$$

$$= \begin{cases} \frac{m}{n} \sum_{i'=1}^{k} E(v_{S,i,i'}^2) \sigma_{S,i'}^2 = m & \text{if } i \in S \\ \frac{m}{n} \sum_{i'=1}^{n} E(v_{S^c,i-k,i'}^2) \sigma_{S^c,i'}^2 = m & \text{if } i \in S^c \end{cases}. \tag{152}$$

Moreover, if $V_S, V_{S^c}$ are fixed, then the rows of $X$ are jointly Gaussian and

$$E\left(X^{m\top} X^m\right) = \frac{1}{n} \left[ \Sigma_S V_S^\top, \Sigma_{S^c} V_{S^c}^\top \right]^\top E\left(W^{m\top} W^m\right) \left[ \Sigma_S V_S^\top, \Sigma_{S^c} V_{S^c}^\top \right] \tag{153}$$

$$= \frac{m}{n} \left[ \Sigma_S V_S^\top, \Sigma_{S^c} V_{S^c}^\top \right]^\top \left[ \Sigma_S V_S^\top, \Sigma_{S^c} V_{S^c}^\top \right]. \tag{154}$$

So if $m = n$, the covariance matches empirical covariance of $X$ constructed in Section 5.1 with $V_S, V_{S^c}$ fixed. The standard Lasso application considers problems in which the noise vector has fixed variance: $w \sim \mathcal{N}(0, \sigma^2 I_{n \times n})$. In the next section we let the variance grow as $\sigma^2 m / n$ (i.e., we use noise vectors $w^m \sim \mathcal{N}(0, (\sigma^2 m / n) I_{m \times m})$) and see how the induced ratio of penalty parameter bounds behaves as $m \to \infty$. Growing the number of observations and noise variance simultaneously ensures that the problem doesn't become too easy.

## 4.3 Convergence of bounds ratios

For some fixed $V_S$, $V_{S^c}$, $\Sigma_S$, $\Sigma_{S^c}$, and $\beta^*$ generate the following two independent Lasso problems.

$$y = X\beta^* + w \qquad X = U\left[\Sigma_S V_S^\top, \Sigma_{S^c} V_{S^c}^\top\right] \qquad w \sim \mathcal{N}(0, \sigma^2 I_{n \times n}) \tag{155}$$

$$y^m = X^m \beta^* + w^m \qquad X^m = \frac{1}{\sqrt{n}} W^m \left[\Sigma_S V_S^\top, \Sigma_{S^c} V_{S^c}^\top\right] \quad w^m \sim \mathcal{N}\left(0, \frac{\sigma^2 m}{n} I_{m \times m}\right), \tag{156}$$

where $U$ is a randomly chosen $n \times n$ orthonormal basis, $W^m$ is a random $m \times n$ Gaussian ensemble, and the noise vectors $w$ and $w^m$ are independent. Now, let $\lambda_u/\lambda_l$ be the ratio of penalty parameter bounds induced by Lemma 1 for the orthonormal construction in Eq. (155) and $\lambda_u^m/\lambda_l^m$ the ratio of penalty parameter bounds for the Gaussian construction in Eq. (156). We will show the following.

**Theorem 4.** *Let $V_S$, $V_{S^c}$, $\Sigma_S$, $\Sigma_{S^c}$ and $\beta^*$ be fixed. If the conditions of Lemma 1 hold for $X, \beta^*$, then for $m$ large enough they will hold for $X^m, \beta^*$. Furthermore, as $m \to \infty$*

$$\frac{\lambda_u^m}{\lambda_l^m} \xrightarrow{d} \frac{\lambda_u}{\lambda_l}, \tag{157}$$

*where the stochasticity on the left is due to $W^m, w^m$ and on the right is due to $w$.*

*Proof.* Let the variables introduced by Lemma 1 for the orthogonal model in Eq. (155) be $\lambda$, $\lambda_l, \lambda_u, \epsilon_i, \gamma_i, \mu_j$ and $\eta_j$. Let the corresponding variables for the Gaussian model of Eq. (156) be $\lambda^m, \lambda_l^m, \lambda_u^m, \epsilon_i^m, \gamma_i^m, \mu_j^m$ and $\eta_j^m$. Similarly, let the counterparts to $X_S$ and $X_j$ be $X_S^m$ and $X_j^m$.

Since we assumed that $V_S, V_{S^c}, \Sigma_S, \Sigma_{S^c}, \beta^*$ are fixed, we first show that $\gamma_i^m$ and $\mu_j^m$ converge to the constants $\gamma_i, \mu_j$. Using the Strong Law of Large Numbers and the Continuous Mapping Theorem,

$$\lim_{m \to \infty} \frac{1}{m} X^{m\top} X^m = \lim_{m \to \infty} \frac{1}{mn} \left[\Sigma_S V_S^\top, \Sigma_{S^c} V_{S^c}^\top\right]^\top W^{m\top} W^m \left[\Sigma_S V_S^\top, \Sigma_{S^c} V_{S^c}^\top\right] \tag{158}$$

$$\stackrel{a.s.}{=} \frac{1}{n} X^\top X \tag{159}$$

This means that all inner products of columns of $X^m/\sqrt{m}$ converge. Then, assuming the conditions of Lemma 1 hold,

$$\lim_{m \to \infty} \gamma_i^m = \lim_{m \to \infty} e_i^\top \left(\frac{1}{m} X_S^{m\top} X_S^m\right)^{-1} \mathrm{sgn}(\beta_S^*) \tag{160}$$

$$\stackrel{a.s.}{=} e_i^\top \left(\frac{1}{n} X_S^\top X_S\right)^{-1} \mathrm{sgn}(\beta_S^*) = \gamma_i \tag{161}$$

$$\lim_{m \to \infty} \mu_j^m = \lim_{m \to \infty} X_j^{m\top} X_S^m (X_S^{m\top} X_S^m)^{-1} \mathrm{sgn}(\beta_S^*) \tag{162}$$

$$= \lim_{m \to \infty} \frac{X_j^{m\top} X_S^m}{m} \left(\frac{1}{m} X_S^{m\top} X_S^m\right)^{-1} \mathrm{sgn}(\beta_S^*) \tag{163}$$

$$\stackrel{a.s.}{=} \frac{X_j^\top X_S}{n} \left(\frac{1}{n} X_S^\top X_S\right)^{-1} \mathrm{sgn}(\beta_S^*) = \mu_j \tag{164}$$

Thus, if the conditions of Lemma 1 hold for $X, \beta^*$, there is an $m_0$ so that if $m > m_0$ the conditions are also met by $X^m, \beta^*$. Assume from now on the conditions are met. By Lemma 1, signed support recovery requires that

$$\lambda^m < \lambda_u^m = \min_{i \in S} \left|\frac{\beta_i^* + \epsilon_i^m}{\gamma_i^m}\right|_+ \tag{165}$$

$$\lambda^m > \lambda_l^m = \max_{j \in S^c} \frac{\eta_j^m}{\left(2[\![\eta_j^m > 0]\!] - 1\right) - \mu_j^m}. \tag{166}$$

We will show that $\lambda_u^m/\lambda_l^m \xrightarrow{d} \lambda_u/\lambda_l$, where the randomness on the left hand side is due to $W^m, w^m$ and the randomness in the right limit is due to the noise $w$ in the $\epsilon_i$ and $\eta_j$. To show this convergence,

observe that we can (with probability 1) write $\lambda_u/\lambda_l$ as a continuous function of $\beta_i^*, \epsilon_i, \gamma_i, i \in S, \eta_j, \mu_j, j \in S^c$, since we have that $\gamma_i > 0, \mu_j \in (-1, +1)$, and $\mathbb{P}(\max_j \eta_j = 0) = 0$ if $\sigma^2 > 0$[3]. By the Continuous Mapping Theorem, convergence in distribution of $\lambda_u^m/\lambda_l^m$ could then be guaranteed if we had the following joint convergence in distribution

$$\begin{bmatrix} \{\epsilon_i^m\}_{i \in S} \\ \{\gamma_i^m\}_{i \in S} \\ \{\eta_j^m\}_{j \in S^c} \\ \{\mu_j^m\}_{j \in S^c} \end{bmatrix} \xrightarrow{d} \begin{bmatrix} \{\epsilon_i\}_{i \in S} \\ \{\gamma_i\}_{i \in S} \\ \{\eta_j\}_{j \in S^c} \\ \{\mu_j\}_{j \in S^c} \end{bmatrix}. \tag{167}$$

Because $\mu_j^m$ and $\gamma_i^m$ converge to constants $\mu_j, \gamma_i$, it remains to be shown that

$$\begin{bmatrix} \{\epsilon_i^m\}_{i \in S} \\ \{\eta_j^m\}_{j \in S^c} \end{bmatrix} \xrightarrow{d} \begin{bmatrix} \{\epsilon_i\}_{i \in S} \\ \{\eta_j\}_{j \in S^c} \end{bmatrix}. \tag{168}$$

To simplify notation, we will show only the marginal convergence, letting it be understood that the argument holds jointly. Using the Strong Law of Large Numbers and Slutsky's Lemma,

$$\lim_{m \to \infty} \epsilon_i^m = \lim_{m \to \infty} e_i^\top \left(\frac{1}{m} X_S^{m\top} X_S^m\right)^{-1} X_S^{m\top} \frac{w^m}{m} \tag{169}$$

$$\stackrel{d}{=} \lim_{m \to \infty} e_i^\top \left(\frac{1}{n} X_S^\top X_S\right)^{-1} X_S^{m\top} \frac{w^m}{m} \tag{170}$$

$$\stackrel{d}{=} \lim_{m \to \infty} \frac{1}{\sqrt{n}} e_i^\top \left(\frac{1}{n} X_S^\top X_S\right)^{-1} V_S \Sigma_S^\top W^{m\top} \frac{w^m}{m} \tag{171}$$

$$\lim_{m \to \infty} \eta_j^m = \lim_{m \to \infty} X_j^{m\top} \left(I_{m \times m} - X_S^m (X_S^{m\top} X_S^m)^{-1} X_S^{m\top}\right) \frac{w^m}{m} \tag{172}$$

$$\stackrel{d}{=} \lim_{m \to \infty} X_j^{m\top} \left(I_{m \times m} - \frac{1}{m} X_S^m \left(\frac{1}{m} X_S^{m\top} X_S^m\right)^{-1} X_S^{m\top}\right) \frac{w^m}{m} \tag{173}$$

$$\stackrel{d}{=} \lim_{m \to \infty} X_j^{m\top} \left(I_{m \times m} - \frac{1}{m} X_S^m \left(\frac{1}{n} X_S^\top X_S\right)^{-1} X_S^{m\top}\right) \frac{w^m}{m} \tag{174}$$

$$\stackrel{d}{=} \lim_{m \to \infty} X_j^{m\top} \left(I_{m \times m} - \frac{1}{mn} W^m \Sigma_S V_S^\top \left(\frac{1}{n} V_S \Sigma_S^\top \Sigma_S V_S^\top\right)^{-1} V_S \Sigma_S^\top W^{m\top}\right) \frac{w^m}{m} \tag{175}$$

$$\stackrel{d}{=} \lim_{m \to \infty} X_j^{m\top} \left(I_{m \times m} - \frac{1}{m} W^m \Sigma_S \left(\Sigma_S^\top \Sigma_S\right)^{-1} \Sigma_S^\top W^{m\top}\right) \frac{w^m}{m} \tag{176}$$

$$\stackrel{d}{=} \lim_{m \to \infty} \frac{1}{\sqrt{n}} v_{S^c, j-k, \cdot} \Sigma_{S^c}^\top W^{m\top} \left(I_{m \times m} - \frac{1}{m} W_S^m W_S^{m\top}\right) \frac{w^m}{m} \tag{177}$$

$$\stackrel{d}{=} \lim_{m \to \infty} \frac{1}{\sqrt{n}} v_{S^c, j-k, \cdot} \Sigma_{S^c}^\top \left(W^{m\top} - \frac{1}{m} W^{m\top} W_S^m W_S^{m\top}\right) \frac{w^m}{m} \tag{178}$$

$$\stackrel{d}{=} \lim_{m \to \infty} \frac{1}{\sqrt{n}} v_{S^c, j-k, \cdot} \Sigma_{S^c}^\top \left(W^{m\top} - \begin{bmatrix} I_{k \times k} \\ 0 \end{bmatrix} W_S^{m\top}\right) \frac{w^m}{m} \tag{179}$$

$$\stackrel{d}{=} \lim_{m \to \infty} \frac{1}{\sqrt{n}} \left[v_{S^c, j-k, \cdot} \Sigma_{S^c}^\top\right]_{(k+1:n)} W_{S^c}^{m\top} \frac{w^m}{m} \tag{180}$$

Observe that since $V_S, V_{S^c}, \Sigma_S, \Sigma_{S^c}$ are fixed, the joint limit distribution of $\left[\{\epsilon_i^m\}_{i \in S}, \{\eta_j^m\}_{j \in S^c}\right]$ is determined by the limit distribution of the *shared* random variable $W^{m\top} w^m/m$. The following lemma allows us to exploit this

**Lemma 4.** *Let $U$ be a (possibly random) $n \times n$ orthonormal matrix and $w \sim \mathcal{N}(0, \sigma^2 I_{n \times n})$. Then*

$$W^{m\top} \frac{w^m}{m} \xrightarrow{d} U^\top \frac{w}{\sqrt{n}}, \tag{181}$$

*Proof.* We show that for an independent $z \sim \mathcal{N}(0, \sigma^2 I_{m \times m})$

$$W^{m \top} \frac{w^m}{m} \xrightarrow{d} \lim_{m \to \infty} W^{m \top} \frac{w^m}{m} \overset{d}{=} \lim_{m \to \infty} W^{m \top} \frac{z}{\sqrt{mn}} \overset{d}{=} \lim_{m \to \infty} W^{m \top} \frac{\sigma z}{\|z\|_2 \sqrt{n}} \overset{d}{=} U^\top \frac{w}{\sqrt{n}} \quad (182)$$

By simple application of the Central Limit Theorem to $W^{m \top} z / \sqrt{m}$ we see that the marginals of the third random variable are Gaussian. To clarify the dependency structure between the variables, we have further modified the statement by explicitly normalizing $z$ on the right. We can do this using Slutsky's Lemma, because by the Strong Law of Large Numbers $\|z\|_2 / \sqrt{m} \overset{a.s.}{\to} \sigma$. Now, since the elements of $W^m$ are independent standard Gaussians, and $z$ has been normalized to unit length, the limit distribution on the right consists of independent zero-mean Gaussians with variance $\sigma^2 / n$. $\square$

Because $V_S, V_{S^c}, \Sigma_S, \Sigma_{S^c}$ are fixed, we can use Lemma 4 to conclude that jointly

$$\begin{bmatrix} \{\epsilon_i^m\}_{i \in S} \\ \{\eta_j^m\}_{j \in S^c} \end{bmatrix} \xrightarrow{d} \begin{bmatrix} \left\{ e_i^\top \left( \frac{1}{n} X_S^\top X_S \right)^{-1} V_S \Sigma_S^\top U^\top \frac{w}{n} \right\}_{i \in S} \\ \left\{ \left[ v_{S^c, j-k, \cdot} \Sigma_{S^c}^\top \right]_{(k+1:n)} U_{S^c}^\top \frac{w}{n} \right\}_{j \in S^c} \end{bmatrix} \overset{d}{=} \begin{bmatrix} \{\epsilon_i\}_{i \in S} \\ \{\eta_j\}_{j \in S^c} \end{bmatrix}. \quad (183)$$

Finally, an application of the Continuous Mapping Theorem to $\epsilon_i^m, \gamma_i^m, \eta_j^m, \mu_j^m$ then establishes that

$$\frac{\lambda_u^m}{\lambda_l^m} \xrightarrow{d} \frac{\lambda_u}{\lambda_l}. \quad (184)$$

$\square$

## Footnotes

[1]For ease of presentation, we let the $d_i$ be distinct.

[2]Note that they switch $V$ with $U$ relative to our notation.

[3]To see this, note that $\left[v_{S^c, j-k, \cdot} \Sigma_{S^c}^\top\right]_{(k+1:n)}$ cannot be zero for all $j \in S^c$.