[Reviews · NeurIPS 2013]

Submitted by Assigned_Reviewer_5

QUALITY:
This paper show some theoretical results concerning preconditioning LASSO algorithms. In these procedures, the data is linearly projected or transformed into a subspace of the data before applying the usual LASSO variable selection procedure on the transformed variables. The papers shed light on the advantages or disadvantages of doing so. However, there is no practical application where to see how to apply their methodology.


ORIGINALITY:
The main result is that by appropriately transforming the data the success of LASSO depends less on the choice of the penalty parameter. That is, there is a wider range of penalty parameter values that give rise to correct variable selection. The authors also show that a bad choice of preconditioning may lead to the failure of LASSO. Their theoretical results allow the author to suggest a new preconditioning method. The authors corroborate their theoretical results with some simulations. They also suggest a probabilistic framework for future study of preconditioning LASSO methods.


CLARITY:
The paper is a bit difficult to follow without reading back and forth. Figure 2 is unintelligible. The axis and the legend cannot be read. The authors should enlarge the figures. What's the meaning of [[ eta_j > 0]] in the formulas for the thresholds?


SIGNIFICANCE:
The results are interesting, but since the bounds found in the theoretical results are random and dependent on unknown variables, it is not clear how to use them in practice or if they can be used in practice.
Summary: The paper shows that preconditioning LASSO may improve variable selection in the sense that the choice of the LASSO penalty parameter becomes less critical. The result is interesting, but it is not clear how it will improve variable selection in practice.

Submitted by Assigned_Reviewer_8

In this paper, the authors analyze a particular class of lasso variants which they call preconditioned lasso algorithms. In these algorithms, both the design matrix and the response are pre-multiplied by some arbitrary matrices. Some specific choices of these matrices are claimed to improve the feature selection capacity of the lasso. However, these claims have so far been insufficiently verified. The main challenge to theoretical analysis here is that the tuning parameters for the preconditioned set up and the "vanilla" set up may no longer be easy to match, and so it is a challenge to show results for particular choices of the tuning parameter \lambda.

The authors focus on the signed support recovery properties of the lasso for linear regression, that is, if the lasso can correctly recover, for each feature, whether its linear coefficient is positive, negative, or identically zero. They study two preconditioning techniques that are based on singular value decompositions of the design matrix. The main tool presented here is to find conditions under which sign support recovery is possible, and comparing these conditions under preconditioned and non-preconditioned settings. Specifically, they prove two theorems related to the penalty parameters, demonstrating that the bounds on a particular choice of penalty allows for signed support recovery are related between the vanilla and preconditioned technique. Following this, the authors a third theorem for a third preconditioning technique following an exposition of the generative lasso model based on the SVD.

Preconditioned lasso problems are not a major topic of research, so the significance of this paper relies on it providing a major reason to use preconditioning. The authors further limit themselves to linear regression lasso, and to support recovery. The technique of bounding the penalty term is original, but its usefulness is unclear from the text.

The paper is fairly readable, but the main results and points are not immediately clear. The authors should state more clearly how and why their theorems may or may not demonstrate signed support recovery. Further, it is not clear how preconditioning affects the difficulty of the signed support recovery problem (and if this is the same as their result anyway?). In any case, the magnitude of the performance gain (if any) is left ambiguous. The reader should be able to more clearly understand what properties have been exposed of the three preconditioned estimators. Again, as preconditioned approaches are not standard, and their advantages are poorly defined, the theoretical results of this paper seem insufficient to justify their use and therefore elevate their impact.

EDIT after rebuttal: in light of the discussions following the rebuttal period, I am more convinced that the results in this paper are useful and novel. I still feel overall that the work is addressing a somewhat obtuse topic, but I appreciate qualities of the work more fully after further reading. My main issues are reasonable topics of future work, and should not block this work. Specifically: the use of ratios of penalty parameters, outside of the degenerate cases (infeasible bounds, infinite or null bounds) should be investigated (is there any gain from a wider "correct" interval, even when scaled by the maximal penalty)? Performance issues beyond existence of a suitable penalty parameter should also be discussed.
Summary: This paper focused on the analysis of preconditioned lasso problems, which are rather a obscure set of techniques whose advantages are not well understood. The authors claim to contribute to clearing this up, but their results are difficult to interpret clearly.

Submitted by Assigned_Reviewer_9

Summary:

This paper introduces and develops an interesting framework for various methods for preconditioning the Lasso. Their framework gets around a difficulty of deciding which tuning parameter value to look at via asking which method would be easiest to tune in finite sample (as measured by how large the range of good parameters is for each method).

Quality & Originality:

This paper addresses a well-motivated problem in a precise and complete (given the page constraints) way. Their framework makes intuitive sense and then they show how it may be fruitfully applied to understanding the preconditioned lasso(s). The work is thoughtful and original. The authors provide empirical support in Figures 1 and 2 for their theoretical findings, supporting that this work was conducted in a thorough, careful manner.

Significance:

The authors demonstrate how their framework allows them to make precise, insightful statements about three existing approaches to preconditioning the lasso. This suggests that the framework is quite significant, especially since it is easy to see that it may be used even in more general problems than preconditioning.

Clarity:

This paper's writing was excellent. The introduction describes the problem extremely well. Proofs (which, of course, are relegated to supplementary materials) are described at a high-level in the text, a practice that really helps the reader understand what is happening (whereas many papers, unfortunately, hide away the main ideas leaving the reader only aware of the theorem statements but with no clue where they came from). The manuscript was clearly very carefully proofread! This reviewer wishes more papers he encountered were written this well.

Comments:
- Lemma 1: can you give some intuition for what goes wrong when lambda is either too small or too big? In particular, can you say why the expressions in (10) have the forms they do in the context of this intuition?
- Line 343: "on" should be "in"?
Summary: An interesting theoretical framework is developed for comparing various preconditioned lasso proposals. The authors, in an extremely well-written and carefully conducted paper, demonstrate that their framework can lead to a better understanding of how these various proposals compare to the lasso and to each other.

Submitted by Assigned_Reviewer_10

The paper addresses preconditioned lasso estimators, and studies their utility from a non-asymptotic, theoretical framework. Their main result is a set of bounds (lower and upper) on lambda that are necessary and sufficient for exact support recovery---they then apply this result to study the effects of different preconditioning methods. I like the general perspective of the paper (though I think the authors could have given a better motivation for their approach). The work is generally well-written, and as far as I can tell, novel. I do have two major comments, given below. After that, I give several more minor comments.

==========
Major comments:

- Lemma 1 seems like a novel and interesting result. But at the same time, I'm suspicious of the "if and only if" here. For this result to be true, it would need to hold that \lambda_l and \lambda_u, as you've defined them, are knots of the lasso solution path. From the LARS algorithm (homotopy algorithm) for the lasso path, we know that the signed support of the lasso solution is piecewise constant, and only changes at data-dependent knots \lambda_1 \geq \lambda_2 \geq \ldots \lambda_r. Hence it either \lambda_l or \lambda_u did not coincidence with one of these knots, then we could slide it to the closest knot (slide to the left for \lambda_l, and to the right for \lambda_u), and we would know that the constructed support wouldn't have changed.

Generally, the formulae for these knots, \lambda_1 \geq \lambda_2 \geq \ldots \geq \lambda_r, are complicated since they depend on the constructed support \hat{S} and signs of the lasso solution along the lasso solution path. You've written down formulae for \lambda_l and \lambda_u that depend on S, not \hat{S}, so at a high level, I don't understand how this can be right.

But at a lower level, skimming through the proof of your Lemma 1 in the supplementary material, I don't see any mistakes. Some comments/clarification on my concerns here would be very helpful.

UPDATE: I now believe that this result is correct. Given the assumed conditions, we have \hat{S}=S (perfect recovery) so my initial intuitive concerns are remedied. Furthermore, what you have written down are actually knots in the lasso path, i.e., these can be shown to coincide with "hitting times", the values of lambda at which variables enter the model.

- Section 3.2: It seems that the ratios of the upper and lower lambda bounds, to compare the usual and preconditioned lasso estimates, is not the right metric---instead you would want to compare their relative widths: ( \lambda_u - \lambda_l ) / \lambda_max, where \lambda_max is the smallest value of \lambda for which all components of the estimate are zero. This has a closed form: \lambda_max = \|X^T y\|_\infty, i.e., the largest absolute component of X^T y. Clearly \lambda_max is going to be different between the two problems (as X becomes P_X X and y becomes P_y y for the preconditioned lasso).

A simple example shows that using the ratios can give somewhat undesirable answers: if \lambda_u = 4 and \lambda_l = 2 for the lasso, and \lambda_u = 2 and \lambda_l = 1 for the preconditioned lasso, then your metric would say they perform about the same. But if in this case \lambda_max = 10 for the lasso and \lambda_max = 3 for the preconditioned lasso, then the story is very different: with respect to the full range of effective regularization parameter values, only 1/5 of this range recovers the support for the lasso, and 1/3 of it recovers the support for the preconditioned lasso.

==========
Minor comments:

- Page 1, line 53: I think you mean sufficient (not necessary) for support recovery

- Page 2, line 80: I'm not really sure what you mean by "asymptotic analyses ... provide limited theoretical insight". Of course, an asymptotic analysis is entirely theoretical, and so yes, it does provide such insight. You mean something like satisfactory insight?
- In motivating your approach on page 2, it seems to me like your method is actually closely aligned with comparing the entire regularization paths of (2) and (4), as done empirically in the reference [4]. You are basically answering the question: "does the regularization path contain the true model, at any point along the path?" But you are doing this in an efficient way by deriving upper and lower bounds on lambda.

- Page 3, line 119: It seems strange to take the q *smallest* singular vectors of U (by this, I assume you mean the singular vector corresponding to the smallest singular values). Can you comment on this? This is especially relevant in Theorem 1, where you impose the condition span(U_S) \subseteq span(U_A). With this construction of A, we would probably not at all expect the sets S and A to be close, so this condition seems quite unreasonable ...

- I'm not sure why you don't present the deterministic result for JR (only one based on a generative model)? For completeness in comparing the preconditioning methods, this would have been helpful.

- Equation (21) is confusing: I think I know what you mean here, but this equation is somewhat confusing without seeing context for \hat{\Sigma} (just calling \hat{\Sigma} "unconstrained spectra" is not that informative) ...

- I'm confused about the setup in 5.3. Why have you changed the number of observations to m, and what does it mean for m \rightarrow \infty while n stays fixed?
Summary: The authors study preconditioning in lasso estimators. They derive lower and upper bounds on lambda that ensure exacts support recovery in the lasso estimator, and use these bounds to evaluate different preconditioning methods. I generally liked the ideas and approaches in the paper. But I had some fairly major issues with it (described in detail the comments section).
Author Feedback

Author rebuttal: We thank the reviewers for their comments and will make the necessary
changes to our paper. We answer some specific questions below.

R8

We apologize for not explaining in greater depth in the main
text, but variable selection in \ell_0 norm is a classical NP-hard
problem whose solution is desperately needed in all of the data-driven
sciences. To date, the Lasso is the most computationally tractable
approximation to this goal. Correlated variables are a perennial
problem for the Lasso and frequently lead to systematic failures. This
deficiency has motivated the use of Preconditioned Lasso
algorithms. Prior to our work, the performance gain of Preconditioned
Lasso algorithms could be evaluated only empirically or
asymptotically. Our theory is useful in making these comparisons on
finite datasets. Furthermore, we view our framework as a stepping
stone towards building improved (possibly iterative) algorithms for
approximating the true \ell_0 solution.

R5

The Preconditioned Lasso algorithms we analyzed (HJ, PBHT, JR) were
already applied to a number of practical applications by the original
authors. Our goal was to develop and apply a new framework in order to
theoretically compare them. We demonstrate this framework in Theorems
1--4. We apologize for not fully explaining our notation; the double
square brackets denote a binary number that is 1 iff the statement
inside the brackets is true. In keeping with other theoretical
developments (e.g. [1]), our theory assumes certain unknown
quantities. Despite this, our theory gives intuition into when
algorithms succeed or fail and allows us to suggest changes to
existing preconditioning algorithms.

R9

We appreciate the reviewer's positive comments that our framework can
lead to a better understanding of Preconditioned Lasso variants. We
also hope that the theory can act as a useful tool for developing more
advanced algorithms. We will augment the paper and build more
intuition for the interpretation of Lemma 1.

R10

Lemma 1 of [10] gives necessary and sufficient conditions to show that
an arbitrary vector \hat \beta (which has support \hat S) is optimal,
given a particular \lambda. The PDW construction used for Lemma 2 of
[10] in effect checks if any vector with the true signed support could
be a solution to the Lasso problem. To do this, Lemma 2 constructs a
potential solution \hat \beta with correct support S and then checks
if this vector is a unique, optimal solution for the choice of \lambda
and if it also has the correct signed support. If it does, then
running Lasso under suitable conditions with the same \lambda would
recover the true signed support. With this logic, the Lemma does not
need to mention \hat{S} to certify signed support recovery.

Other comparison metrics involving \lambda_l, \lambda_u could be
considered, and the reviewer's suggestion could be a useful
addition. However, the algebraic degeneracy outlined by the reviewer
would require relatively specific dependence of \lambda_{max} on
\lambda_u, \lambda_l. We feel that the ratio statistic is an intuitive
and useful first step when focussing on signed support
recovery. Future work will consider the alternatives in greater depth.

Section 1.1 in the supplementary material shows that HJ can be
rewritten as a Preconditioned Lasso algorithm that uses the q left
singular vectors with smallest singular values. Huang and Jojic
proposed their algorithm HJ as a technique to avoid selecting highly
correlated proxy variables that could be introduced by certain latent
variable arrangements. They argue that in these cases removing a
small number of high-energy directions of X can be useful (see Section
4 of [4]). Theorem 1 shows that HJ is useful for improving signed
support recovery of Lasso if U_A is properly estimated. Future work will need to investigate theoretically under what practical circumstances this can be expected to be case.

By "unconstrained" spectrum matrices we mean unnormalized spectrum
matrices. That is, the matrices \hat \Sigma_S, \hat \Sigma_{S^c} can
have arbitrary positive elements on the diagonal.

In 5.3 we consider what happens when we replace the orthogonal left
basis U of Eq. (20) with a general Gaussian basis W in Eq. (26). While
a direct comparison of this model with the generative model (20)
cannot be made, we show that if the number m of rows in W grows with
the noise variance, the induced ratio of upper and lower penalty
parameter bounds for (26) can be compared with that of the generative
model (20). To do this, we show in the supplementary material (Section
4.4) that if m is large, the columns of W are almost orthogonal
(i.e. (1/m) W' W -> I, see Eqs. (155), (156)) and that for large m,
the inner product (1/m) W' w^m (where w^m is a noise vector) behaves
as a Gaussian vector due to the Central Limit Theorem (see Lemma
3). For both of these limits we need m -> infinity, while n is fixed.